

**Review Article**
**Antarctica's internal architecture:**
**Towards a radiostratigraphically-informed age–depth model of the**
**Antarctic ice sheets**
Robert G. Bingham[1]\*, Julien A. Bodart[2,3]\*, Marie G. P. Cavitte[4]\*, Ailsa Chung[5]\*, Rebecca J. Sanderson[6]\*,
Johannes C. R. Sutter[2,3]\*, Olaf Eisen[7,8], Nanna B. Karlsson[9], Joseph A. MacGregor[10], Neil Ross[6], Duncan A.
Young[11], David W. Ashmore[12], Andreas Born[13,14], Winnie Chu[15], Xiangbin Cui[16], Reinhard Drews[17], Steven
Franke[7,17], Vikram Goel[18], John W. Goodge[19,20], A. Clara J. Henry[21], Antoine Hermant[2,3], Benjamin H. Hills[22],
Nicholas Holschuh[23], Michelle R. Koutnik[24], Gwendolyn J.-M. C. Leysinger Vieli[25], Emma J. MacKie[26], Elisa
Mantelli[7,27], Carlos Martín[28], Felix S. L. Ng[29], Falk M. Oraschewski[17], Felipe Napoleoni[1], Frédéric Parrenin[5],
Sergey V. Popov[30,31], Therese Rieckh[13,14], Rebecca Schlegel[17], Dustin M. Schroeder[32,33], Martin J. Siegert[34],
Xueyuan Tang[16,,35], Thomas O. Teisberg[33], Kate Winter[36], Shuai Yan[11], Harry Davis[1], Christine F. Dow[37,38], Tyler
J. Fudge[24], Tom A. Jordan[28], Bernd Kulessa[39], Kenichi Matsuoka[40], Clara J. Nyqvist[1], Maryam
Rahnemoonfar[41,42], Matthew R. Siegfried[22], Shivangini Singh[11,43], Vjeran Višnjević[2,3], Rodrigo Zamora[44],
Alexandra Zuhr[17]
\* These authors contributed equally to this work.
*Correspondence to:* Robert G. Bingham (r.bingham@ed.ac.uk)
**Affiliations**
[1] School of GeoSciences, University of Edinburgh, UK
[2] Climate and Environmental Physics, Physics Institute, University of Bern, Switzerland
[3] Oeschger Centre for Climate Change Research, University of Bern, Switzerland
[4] Earth and Life Institute, Université catholique de Louvain (UCLouvain), Belgium
[5] Univ. Grenoble Alpes, CNRS, INRAE, IRD, Grenoble INP, IGE, 38000 Grenoble, France
[6] School of Geography, Politics and Sociology, Newcastle University, UK
[7] Alfred Wegener Institute Helmholtz Centre for Polar and Marine Research, Bremerhaven, Germany
[8] Department of Geosciences, University of Bremen, Germany
[9] Department of Glaciology and Climate, Geological Survey of Denmark and Greenland (GEUS), Copenhagen,
Denmark
[10] Cryospheric Sciences Laboratory, NASA Goddard Space Flight Centre, Greenbelt, Maryland, USA



[11] University of Texas Institute for Geophysics, Jackson School of Geosciences, University of Texas at Austin,
Texas, USA
[12] Met Office, FitzRoy Road, Exeter, UK
[13] Department of Earth Science, University of Bergen, Norway
[14] Bjerknes Centre for Climate Research, University of Bergen, Norway
[15] School of Earth and Atmospheric Science, Georgia Institute of Technology, Atlanta, Georgia, USA
[16] Polar Research Institute of China, Shanghai, China
[17] Department of Geosciences, University of Tübingen, Germany
[18] National Centre for Polar and Ocean Research (NCPOR), Ministry of Earth Sciences, Vasco da Gama, Goa,
India
[19] Department of Earth and Environmental Sciences, University of Minnesota Duluth, Duluth, Minnesota, USA
[20] Planetary Science Institute, Tucson, Arizona, USA
[21] Department of Mathematics, Stockholm University, Sweden
[22] Department of Geophysics, Colorado School of Mines, Golden, Colorado, USA
[23] Department of Geology, Amherst College, Massachusetts, USA
[24] Department of Earth and Space Sciences, University of Washington, Seattle, Washington, USA
[25] Department of Geography, University of Zurich, Switzerland
[26] Department of Geological Sciences, University of Florida, Gainesville, Florida, USA
[27] Department of Earth and Environmental Sciences, Ludwig-Maximillians-Universität, Munich, Germany
[28] British Antarctic Survey, Cambridge, UK
[29] Department of Geography, University of Sheffield, Sheffield, UK
[30] St Petersburg State University, Russia
[31] Polar Marine Geosurvey Expedition, St. Petersburg, Russia
[32] Department of Geophysics, Stanford University, California, USA
[33] Department of Electrical Engineering, Stanford University, California, USA
[34] Tremough House, Penryn Campus, University of Exeter, Cornwall, UK
[35] School of Oceanography, Shanghai Jiao Tong University, China
[36] Department of Geography and Environmental Sciences, Northumbria University, Newcastle, UK



[37] Department of Geography and Environmental Management, University of Waterloo, Ontario, Canada
[38] Department of Applied Mathematics, University of Waterloo, Ontario, Canada
[39] School of Biosciences, Geography and Physics, Swansea University, UK
[40] Norwegian Polar Institute, Tromsø, Norway
[41] Department of Computer Science and Engineering, Lehigh University, Pennsylvania, USA
[42] Department of Civil and Environmental Engineering, Lehigh University, Pennsylvania, USA
[43] Department of Earth and Planetary Sciences, University of Texas at Austin, Austin, Texas, USA
[44] Centro de Estudios Científicos, Valdivia, Chile



**Abstract.** Radio-echo sounding (RES) has revealed an internal architecture within Antarctica's ice sheets that records their depositional, deformational and melting histories. Crucially, spatially-widespread RES-imaged internal-reflecting horizons, tied to ice-core age-depth profiles, can be treated as isochrones that record the age-depth structure across the Antarctic ice sheets. These enable the reconstruction of past climate and ice-dynamical processes on large scales, which are complementary to but more spatially-extensive than commonly used proxy records across Antarctica. We review progress towards building a pan-Antarctic age-depth model from these data by first introducing the relevant RES datasets that have been acquired across Antarctica over the last six decades (focussing specifically on those that detected internal-reflecting horizons), and outlining the processing steps typically undertaken to visualise, trace and date (by intersection with ice cores, or modelling) the RES-imaged isochrones. We summarise the scientific applications to which Antarctica's internal architecture has been applied to date and present a pathway to expanding Antarctic radiostratigraphy across the continent to provide a benchmark for a wider range of investigations: (1) Identification of optimal sites for retrieving new ice-core palaeoclimate records targeting different periods; (2) Reconstruction of surface mass balance on millennial or historical timescales; (3) Estimates of basal melting and geothermal heat flux from radiostratigraphy and comprehensively mapping basal-ice units, to complement inferences from other geophysical and geological methods; (4) Advancing knowledge of volcanic activity and fallout across Antarctica; (5) The refinement of numerical models that leverage radiostratigraphy to tune time-varying accumulation, basal melting and ice flow, firstly to reconstruct past behaviour, and then to reduce uncertainties in projecting future ice-sheet behaviour.



## 1 Introduction

Throughout the Quaternary (2.58 Ma to present), Antarctica's ice cover has waxed and waned, inducing concomitant rises and falls in global sea level on the order of several tens of metres (e.g., Drewry, 1983; Pollard and DeConto, 2009; Dutton et al., 2015). It is critical to understand the rates and drivers of these past oscillations in order to contextualise current observations of persistent and accelerating losses from the contemporary Antarctic ice sheets (e.g., Fox-Kemper et al., 2021; Otosaka et al., 2023) and thereby project as accurately as possible the rates at which future global sea-level rise fuelled by ice melt will occur (e.g., Scambos et al., 2017; Oppenheimer et al., 2019). The evidence for past Antarctic ice-sheet fluctuations has been derived predominantly from sampling sediments deposited offshore around the continent (Escutia et al., 2009; Naish et al., 2009; Cook et al., 2013; Bentley et al., 2014; Gulick et al., 2017; Hillenbrand et al., 2017), dating the exposure history of onshore bedrock and moraine boulders (Brook and Kurz, 1993; Mackintosh et al., 2014; Hillebrand et al., 2021), and by analysing the ice itself recovered from ice-core sites (e.g., EPICA Community Members, 2004; Jouzel et al., 2007; Higgins et al., 2015; WAIS Divide Project Members, 2015; Dome Fuji Ice Core Project Members, 2017; Yan et al., 2021) (see Brook and Buizert, 2018 for an overview). Together, these form the palaeoclimate records that underpin numerical-modelling reconstructions of past and present ice-sheet extents and inform projections of how these may evolve into the future and affect sea-level change (e.g., Gasson et al., 2016; Golledge et al., 2019; DeConto et al., 2021; Pittard et al., 2022). Recovery of further sediment and ice cores around Antarctica to refine these records and projections remains a scientific imperative – and yet these records are intrinsically spatially limited. Radio-echo sounding across Antarctica complements these records by providing *spatially continuous* data that record past and present ice conditions and, by extension, past and present climate conditions, across the ice sheets.

Radio-echo sounding (RES) describes the investigation of the subsurface of ice sheets using electromagnetic waves, and has been conducted from both airborne and ground-based platforms across the Antarctic ice sheets for over 60 years (see reviews by Dowdeswell and Evans, 2004; Bingham and Siegert, 2007; Allen, 2008; Schroeder et al., 2020). Primarily deployed for mapping the ice-sheet bed and thereby measuring ice thickness and thus ice volume, the majority of RES surveys have also imaged numerous englacial features, predominantly internal-reflection horizons (a.k.a. internal or englacial layers), crevasses and rheologically-distinct "basal units" of ice that occur between the more obvious reflections of the ice surface and bed (Fig. 1). For this review, we collectively term all of the Antarctic ice sheets' RES-imaged englacial features its *internal architecture*. We will demonstrate that although great progress has already been made in using some of this resource to elucidate ice and climate history, Antarctica's internal architecture has yet to be exploited to its full potential in refining our understanding of past, present and future ice-sheet behaviour.

In Greenland, a comprehensive archive of internal architecture has already been assembled (see MacGregor et al., 2015a), facilitating the ice-sheet-wide reconstruction of past accumulation and dynamics, to improve



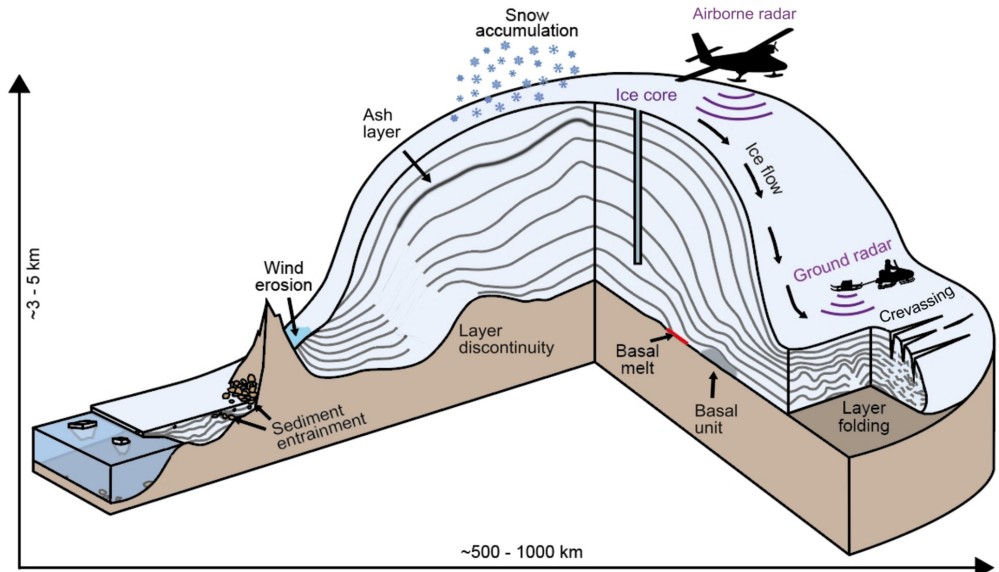

**Figure 1.** Schematic illustration of Antarctica's internal architecture and the key processes governing its structure. Internal-reflection horizons - the ice sheet's "radiostratigraphy" - are represented by grey lines between the surface and bed.


past and future sea-level estimates (MacGregor et al., 2016; Born and Robinson, 2021). However, several
major issues have confounded parallel progress in capturing and applying internal architecture across
Antarctica, including:
1) The Antarctic ice sheets together cover eight times the area of the Greenland Ice Sheet.
2) RES data have been collected, processed and archived by multiple international groups across the Antarctic
ice sheets, and hence are not available in a standardised form across Antarctica.
3) A comprehensive suite of strategies for using internal architecture in numerical ice-sheet models has not
been developed.
4) Much internal architecture in RES data is highly challenging to identify and map with automated methods.
To address these challenges and work collectively towards consistently capturing and utilising Antarctica's
internal architecture, an international community called *AntArchitecture* was formed in 2018. This
community, coordinated via the *Scientific Committee for Antarctic Research* (SCAR), aspires to the ultimate
scientific aim of using Antarctica's internal architecture to deconvolve its ice sheets' histories and thereby
facilitate improved projections of their future behaviour in the face of global climate warming. A first step in
this process, and one of the aims of this review, collectively written by the *AntArchitecture* community, is to
compile the international community's understanding of the present state of the field in terms of available



RES data across the Antarctic ice sheets and their potential applications. Additionally, we seek here to relay
community aspirations to address the aforementioned challenges and position Antarctica's internal
architecture as a valuable resource for improving our understanding of its ice/climate interactions.
We begin with a brief overview of what gives rise to internal architecture in ice, especially the internal-
reflection horizons (hereafter IRHs) that are measured by RES (Sect. 2). We continue by summarising the key
RES datasets acquired across Antarctica that image internal architecture, to contextualise in a single place
the type and quality of information recorded by each institute and survey in the past six decades (Sect. 3). In
Sect. 4, we turn to how RES data have been, and can be, processed to optimise the extraction of internal
architecture and its visualisation; discuss the common methods currently used to characterise and date IRHs;
and finally build an inventory of existing IRH datasets. In Sect. 5, we review how internal architecture has
been used to reconcile ice-core records, calculate changes to past surface mass balance, explore basal
melting in association with subglacial lakes and areas of enhanced geothermal heat flux, and investigate ice-
sheet dynamics and other glaciological questions; and outline how the internal architecture has begun to be
used in in numerical-modelling applications to date. In Sect. 6, we outline a recommended pathway to
building a pan-Antarctic database of Antarctica's internal architecture, and discuss key science deliverables
that can be facilitated by this activity.
**2 Internal architecture in ice sheets**
The most common way in which internal architecture is viewed and assessed is as radargrams, which are
two-dimensional profiles of echo power arrayed in the along-track direction (e.g., Fig. 2). Antarctic
radargrams commonly display clear *radiostratigraphy*, the collective term for the multiple sub-parallel and
closely-spaced IRHs that are seen in radargrams and often, although do not always, broadly follow the shape
of the ice-bed interface (e.g., Fig. 2). IRHs occur as radio-waves propagate down through the ice column and
reflect off any boundary where there is a contrast in the dielectric properties within the ice. The propagation
of radio-waves through snow, firn and ice is controlled by the complex relative permittivities of these
materials, which are functions of density, electrical conductivity, and/or the development of ice-fabric
anisotropy where ice crystals align into a preferential orientation as a result of large englacial stress. Where
contrasts in any of these properties are sufficiently strong and sharp, the incident energy will partition and a
small fraction of it will be reflected back to the RES receiver at or above the ice surface.
In the upper and middle part of the ice column, radiostratigraphy typically arises from (a) density variations,
as snow compacts into ice (as explained in pioneering work by Robin et al. (1969) and Clough (1977)) and (b)
variations in electrical conductivity, as volcanic aerosols present in the air during snow deposition are
incorporated into the firn (Hammer, 1980; Millar, 1981; Millar, 1982). These density- and electrical-
conductivity-derived IRHs are related to snow and ice layers of a specific age buried under subsequent snow



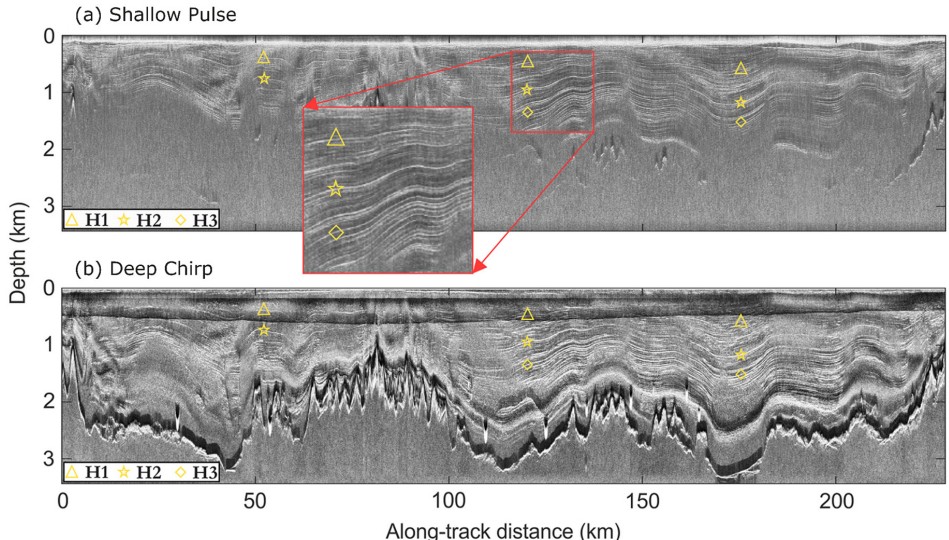

**Figure 2.** Radargrams from Institute Ice Stream, West Antarctica, obtained by the British Antarctic Survey PASIN RES system in (a) pulse (shallow-sounding) and (b) chirp (deep-sounding) radar modes (Frémand et al., 2022), vertically differentiated to accentuate fine detail. Symbols highlight three IRHs found widely across West Antarctica in airborne radar data. The bed reflection (black-white interface) is partially visible in (a) and clearly visible in (b). Figure modified from Ashmore et al. (2020).


accumulation, and thus may be considered isochronous (Hempel et al., 2000; Eisen et al., 2006). Such RES-
imaged isochrones may often represent composites of multiple real horizons in the ice, and their thickness
is dependent on RES-system resolution (Harrison, 1973; Winter et al., 2017). They are often traceable for
considerable distances on RES profiles: some IRHs in the Antarctic and Greenland ice sheets are continuous
for hundreds or even thousands of kilometres (e.g., MacGregor et al., 2015a; Winter et al., 2019a; Ashmore
et al., 2020). For the focus of this review, isochronous reflections arising from density and electrical
conductivity are of significant interest, and IRHs that can be dated at ice cores and traced continuously over
long distances to form a "*dated radiostratigraphy*" are particularly valuable (as explored in-depth in Sect. 4
and 5). There are, however, some cases, especially in the lower part of the ice column, where diachronous
IRHs (i.e. IRHs that cannot be treated as single time markers) may be visualised in radargrams. The most
common such examples are IRHs that are thought to manifest sudden changes in ice-crystal-orientation
fabric that cause anisotropic radio-wave propagation, or cold-warm ice transitions where the pore space on
the warm side is filled with meltwater instead of air (Harrison, 1973; Fujita et al., 1999; Eisen et al., 2007).
Over ice shelves, pervasive IRHs can mark the boundary between atmospherically-derived (meteoric) and
subglacially/submarine-accreted (marine) ice (Holland et al., 2009; Das et al., 2020).
The specular behaviour of IRHs also positions them as ideal targets for repeated observations of vertical
velocity over time, directly tracking the deformation of the ice sheet, via static phase-sensitive repeat



measurements at a point (autonomous phase-sensitive radio-echo sounder, or ApRES; Nicholls et al., 2015)
or from airborne re-flights of transects with coherent RES systems (Castelletti et al., 2021). Although these
methods have been practiced in recent field campaigns (e.g., Hills et al., 2022; Chung et al., 2023; Fudge et
al., 2023), we do not discuss this aspect of radiostratigraphy further in this review, beyond noting that
establishing the distribution of appropriate IRHs could be a valuable component in expedition planning. A
review of static techniques is found in Kingslake et al. (2014), while repeat-pass airborne interferometry of
IRH is an active field of research.
While the imaging and analysis of radiostratigraphy and its application to assessing ice-sheet stability forms
the main focus of this paper, other significant features of internal architecture also convey information that
can be used to help understand current and past ice-sheet processes (as depicted in Fig. 1). These include
basal units which exhibit different dielectric properties to the surrounding ice and may result from ice-folding
due to contrasts in material properties, to accretion, melting due to high rates of geothermal heat flux or
overburden pressure from the ice above, or freeze-on processes taking place at the base of the ice sheet (Bell
et al., 2011; Bell et al., 2014; Bons et al., 2016; Leysinger Vieli et al., 2018; Wrona et al., 2018; Ross et al.,
2020; Franke et al., 2023). Additionally, buried near-surface and basal crevassing imaged by RES systems may
be indicative of past grounding-line evolution or ice-stream stagnation events (Retzlaff et al., 1993; Matsuoka
et al., 2009; Catania et al., 2010; Kingslake et al., 2018; Wearing and Kingslake, 2019). We elaborate further
on these other significant features of internal architecture in Sect. 5.5.
**3 Radio-echo sounding datasets for characterising Antarctica's internal architecture**
Antarctic ice-penetrating RES data have been collected in a series of regional surveys for over six decades. A
broad overview of the history can be gained from the periodic release of maps of subglacial topography, the
first by Drewry (1975) and Drewry (1983; Antarctica Glaciological and Geophysical Folio Sheet 9), and then
through the Bedmap series, now in its third iteration (Frémand et al., 2023; their Fig. 1). However, those
maps outline only where RES data have been used to pick an echo at the ice bed, and crucially do not provide
any information on whether the constituent surveys also captured or recorded any information on internal
architecture. Therefore, we review here specifically which of the RES datasets acquired over Antarctica do
contain, or are likely to contain, useable internal architecture.
The most relevant datasets for characterising internal architecture across Antarctica derive from airborne
RES surveys, as they are the most spatially extensive (extending over thousands of kilometres; Fig. 3), typically
deploy more advanced and more powerful sounders relative to most contemporaneous ground-based
systems, and now commonly employ state-of-the-art processing methods. These qualities favour the
detection of multiple IRHs over wide-ranging regions, resolving IRHs at higher resolution and to greater
depths in the ice sheet. Accordingly, we focus mainly on airborne RES surveys below, in Sect. 3.1 to 3.8



progressing chronologically by order of first Antarctic operations by each main airborne provider, and then
in Sect. 3.9 outlining briefly some additional airborne RES datasets acquired by other groups since airborne
surveying began in the 1960s. Our focus in this review, and hence throughout Sect. 3, is on "deep" RES
datasets, i.e. those that sound the full ice column to several kilometres' depth. Also acquired across many
regions of Antarctica are several additional datasets of "shallow" RES - i.e. that image IRHs in finer detail
down to a few 100 m below the ice surface – which provide complementary resources for work typically
focussed on ice-climate interactions during more recent periods (i.e., the past few hundred years; e.g.,
Medley et al., 2014). To give shallow RES data and applications equal attention to their deeper counterparts
throughout this review would have made the paper unwieldy, but shallow IRHs imaged in RES data certainly
represent another hugely important and rich resource for palaeoclimate modelling and we return to this in
Sect. 6.2.3 when laying out future scientific aspirations.
Complementing the wide-ranging information acquired by airborne RES, several groups have acquired RES
data from vehicles driven along the snow surface. Ground-based RES (described as ground-penetrating radar,
GPR, in some glaciological literature) has typically been deployed to conduct dense surveys around sites of
particular glaciological and geophysical interest, but long exploratory traverses of several hundreds of
kilometres have also been undertaken. Ground-based RES surveys, usually operated with lower frequencies,
benefit from direct coupling to the ice (or snow/firn) surface, so are often particularly effective at mapping
local radiostratigraphy at fine vertical resolution or for deciphering the processes that influence the larger-
scale radiostratigraphy. Perhaps most notably for the purposes of building a pan-Antarctic age-depth model
are those ground-based surveys that can link between two or more large regions, and so in Sect. 3.10 we
outline where such far-ranging surveys have occurred. In parallel with the approach described for airborne
RES data above, here we introduce only the ground-based RES datasets that penetrate through the full ice
thickness.
Two important considerations to introduce before we proceed with introducing where RES data have been
obtained over Antarctica are whether the data were acquired digitally and/or coherently. While the majority
of the datasets discussed here were recorded digitally, RES data acquired before the 1990s were typically
recorded onto analogue tape recorders or film. Very few of these analogue RES datasets have been digitised,
with Schroeder et al. (2019) being a notable exception that has made automated digital interpretation of the
data possible and greatly increased their value for modern analyses. (Karlsson et al. (2024) provides an
equivalent legacy dataset for Greenland.) The use of pre-1990s RES datasets is also challenged by navigational
uncertainties occasioned by their acquisition from before digital navigation systems supported by Global
Navigation Satellite System (GNSS) were fully integrated into survey platforms. By their nature, the analogue
datasets were acquired incoherently, meaning that the RES systems only recorded signal amplitude and not
phase. Until the 2000s, when most airborne RES systems were equipped with GNSS and acquiring data
digitally, most RES systems remained incoherent. Despite this limitation, such systems have successfully



imaged internal architecture, and indeed many ground-based RES systems presently deployed in Antarctica
remain incoherent. The advantage of incoherent ground-based systems is the relative simplicity of operating
and maintaining such RES systems in challenging field conditions. However, with improvements in technology
through the late 1990s/early 2000s, all of the airborne RES operators gradually transitioned to operating
coherent RES systems that detect both returned power and phase, permitting synthetic aperture radar (SAR)
processing of the data (see Sect. 4). This has been crucial for imaging finer details such as low-amplitude
englacial reflections lower in the ice column and across complex terrain that previously was shrouded by
scattering and frequently characterised as echo-free (Hélière et al., 2007; Peters et al., 2007). The overall
progression of RES systems from analogue to digital, from not having digital navigation to navigating with
high-precision GNSS, and from incoherent to coherent RES systems, is depicted in Fig. 3, and introduced in
further selected details below.

### 3.1 Scott Polar Research Institute / National Science Foundation / Technical University of Denmark

From the mid-1960s the UK-based Scott Polar Research Institute (SPRI) began airborne RES surveying across
parts of Antarctica, initially supported logistically by a combination of the British Antarctic Survey (BAS) and
the USA's National Science Foundation (NSF) in reconnaissance flights in the Antarctic Peninsula, and out of
McMurdo and South Pole stations (Swithinbank, 1969; Evans and Smith, 1970; Drewry, 2023). From 1971,
engineers from the Technical University of Denmark (DTU) added antennas designed to operate at 60 MHz
centre frequency for improved reflection of IRHs (Gudmandsen et al., 1975) and thus commenced the earliest
extensive airborne RES campaigns across Antarctica which continued throughout the 1970s (Turchetti et al.,
2008). The SPRI-NSF-DTU surveys profiled >400,000 km across nearly half of the continent, contributing much
of the first iteration of Bedmap (Lythe et al., 2001) across West Antarctica and over East Antarctica between
Wilkes Land, the South Pole and Domes A and C (Fig. 3b). The clarity of IRHs in the 1970s SPRI-NSF-DTU
datasets rivals that sounded in many modern RES surveys, but use of the data is challenging because (1) they
were recorded onto 35-mm optical film and (2) navigation techniques before the use of GNSS were less
precise, leading to several kilometres of positional uncertainties (Schroeder et al., 2019). In the early 2000s,
many of the films were scanned as non-georectified digital images, from which a first archive of
radiostratigraphy across West Antarctica was constructed (Siegert et al., 2005). This seeded many early
applications of radiostratigraphy to glaciological problems across both ice sheets (e.g., Hodgkins et al., 2000;
Siegert and Hodgkins, 2000; Rippin et al., 2003b; Leysinger Vieli et al., 2004; Siegert and Payne, 2004; Siegert
et al., 2004; Bingham et al., 2007; Leysinger Vieli et al., 2011). All those studies acknowledged the inherent
limitations of using analogue data with low positional accuracy. Recently, the SPRI-NSF-DTU data have been
revived by a new finer-resolution digitisation and distribution programme (Schroeder et al., 2019; Schroeder
et al., 2022), which has substantially improved the visibility and accessibility of this wide-ranging
radiostratigraphy. Navigational uncertainties remain, but the radiometric digitisation process offers the







**Figure 3**. (a) Reference map of main Antarctic locations mentioned in this review. (b) to (j) Airborne RES coverage by data provider as discussed through Sect 3.1 to 3.9. Each legend outlines basic details of the provider's RES system by system-name, typical centre frequency, whether the system was incoherent (inc.) or coherent (coh.), whether the acquisition was analogue (an.) or digital (dig.), whether flight navigation used GNSS (assumed not for data collection before 1990), and the date ranges over which a system was used. (k) Coverage of long-range ground-based RES data across Antarctica with potential for extraction of deep internal-reflecting horizons. (l) Combined coverage of all digital RES datasets across Antarctica with potential for contributing to *AntArchitecture*.





prospect of using crossovers with more modern datasets to reconstruct the navigation with improved
accuracy (Teisberg and Schroeder, 2023).

**3.2 Soviet / Russian Antarctic Expedition**

Airborne RES surveying of Antarctica coordinated by the Soviet (later Russian) Antarctic Expedition began in
the mid-1960s. Surveys undertaken with a 60 MHz system, designed primarily to sound the bed but also
capable of imaging IRHs, were conducted between 1967 and 2014, after which all data acquisition was
conducted with a new 130 MHz RES system (Popov, 2020). Throughout the 1980s, systematic surveying was
conducted across large swathes of East Antarctica, extending across Enderby Land and to Vostok Station and
Domes A and F (Popov, 2020). For the early decades of these RES surveys, as for the SPRI-NSF-DTU surveys,
the data were recorded onto film and have a spatial accuracy of several kilometres due to not benefitting
from GNSS navigation; however, they likely contain a rich resource of radiostratigraphy which could be
particularly important because a number of these surveys span approximately one-fifth of East Antarctica
that is otherwise mostly unsurveyed (compare Fig. 3c and 3l). From the 1990s onwards, Russian airborne RES
surveying continued systematically around coastal East Antarctica between ~20°E and 95°E, generally
extending at most 500 km inland to 75°S (Fig. 3c; Popov, 2020; Popov, 2022). A key development for the
ready recovery and future utilisation of radiostratigraphy from these datasets was the switch from analogue
to digital data acquisition that took place in the early 2000s.

**3.3 British Antarctic Survey**

The British Antarctic Survey (BAS) has performed large-scale airborne RES surveys of Antarctica since the
1960s. Until the late 1970s, before which BAS field logistics were run centrally but BAS science was led out
of university research groups, the RES system-development and data analysis were the responsibility of SPRI,
and the RES systems that were deployed were as described in Sect. 3.1. As BAS became more autonomous
from the mid-1970s it transitioned to developing and running its own in-house RES systems, which
progressed in the early 2000s from incoherent to coherent systems (Robin et al., 1977; Corr et al., 2007;
Frémand et al., 2022). Prior to the early 2000s, BAS surveys focused on the Antarctic Peninsula and Filchner-
Ronne Ice Shelf and data were recorded only in analogue form (Fig. 3d). From 2004 onwards, BAS transitioned
to digital data acquisition (Rippin et al., 2003a; Ferraccioli et al., 2005) by developing the 150 MHz, higher-
power, coherent Polarimetric Radar Airborne Science Instrument (PASIN; Corr et al., 2007; Hélière et al., 2007;
Frémand et al., 2022). PASIN was upgraded in the mid-2010s to enable the acquisition of swaths (i.e. wide
strips) of RES data to map the ice-sheet bed (Arenas-Pingarrón et al., 2023). PASIN transmits two waveforms,
a narrow pulse (0.1 µs) for detecting shallow radiostratigraphy in the upper 2 km of the ice column, and a
deep-sounding chirp (4 µs) for detecting deeper radiostratigraphy and the bed (see Fig. 2 for examples of
each). It has been deployed widely across Antarctica (Fig. 3d) and has detected radiostratigraphy across both
West and East Antarctica (Karlsson et al., 2009; Karlsson et al., 2014; Bingham et al., 2015; Winter et al., 2015;



Ashmore et al., 2020; Ross et al., 2020; Bodart et al., 2021; Bodart et al., 2023; Sanderson et al., 2023).
Recently, >450,000 km of PASIN radargrams acquired between 2004 and 2020 were made accessible in open-
access format (Frémand et al., 2022).
**3.4 University of Texas Institute for Geophysics**
The USA-based University of Texas Institute of Geophysics (UTIG) has conducted airborne RES surveys of
Antarctica since the early 1990s, using several generations of systems of increasing sophistication, all with a
centre frequency of 60 MHz (Young et al., 2016). Their earliest surveys, principally of West Antarctica, used
adapted versions of the system used for the SPRI-NSF-DTU surveys and were recorded digitally but
incoherently (Blankenship et al., 2001; Carter et al., 2007). In the early 2000s, UTIG integrated a coherent
RES system (Moussessian et al., 2000) with the DTU radio-frequency hardware to allow high-power coherent
recording, which enabled synthetic-aperture-radar (SAR) processing of acquired data (Peters et al., 2005;
Peters et al., 2007; SAR processing is described in Sect. 4.1). This initial High-Capability Radar Sounder (HiCARS)
system (Blankenship et al., 2017a) was translated to commercially available components (HiCARSII,
Blankenship et al, 2017b) which were incorporated into the subsequent Multifrequency Airborne Radar-
sounder for Full-phase Assessment (MARFA), capable of cross-track interferometry for clutter discrimination
(Castelletti et al., 2017; Scanlan et al., 2020). These systems have successfully detected detailed
radiostratigraphy throughout the Antarctic ice sheets (Fig. 3e) as part of large-scale multi-national campaigns
(e.g., Morse et al., 2002; Carter et al., 2007; Muldoon et al., 2018; Beem et al., 2021) including, from 2008,
across large regions of East Antarctica as part of the ICECAP (Investigating the Cryospheric Evolution of the
Central Antarctic Plate) international consortium (e.g., Young et al., 2011; Wright et al., 2012; Cavitte et al.,
2016; Cavitte et al., 2018) and integrated into NASA's Operation IceBridge (OIB; other airborne RES surveys
by OIB are introduced in Sect. 3.6). RES systems based on the commercial HiCARSII design have been
integrated into the Chinese (Sect. 3.8) and Korean Antarctic programmes, and UTIG is collaborating with
CReSIS (Sect. 3.6) on the mapping of Dome A as part of the US National Science Foundation's Center for
Oldest Ice Exploration (COLDEX).
**3.5 Alfred-Wegener Institute**
The Germany-based Alfred-Wegener Institute (AWI) has performed airborne RES surveys since the mid-1990s
(Steinhage et al., 2001), recording digitally and acquiring a total of ~420,000 km of RES data (Fig. 3f), often as
part of multinational projects. Its primary system until the mid-2010s – the Aero-EMR (Electro-Magnetic
Reflection) instrument – operated around a centre frequency of 150 MHz in a toggle mode that allowed for
short (60 ns; high resolution but low-penetration depth) and long (600 ns; low-resolution but high-
penetration depth) pulses to be transmitted simultaneously (Nixdorf et al., 1999; Eisen et al., 2007). Following
progressive upgrades to the flexibility and sensitivity of its Aero-EMR, AWI began using an improved version
(MCoRDS5) of the CReSIS ultra-wideband RES system (Hale et al., 2016; see Sect. 3.6 below). Antarctic



operations of this newer system have so far operated across Dronning Maud Land using frequencies ranging
from 180-210 and 150-520 MHz, respectively (Franke et al., 2021; Koch et al., 2023; Franke et al., 2024). AWI
RES data (Fig. 3f) have been used extensively to recover radiostratigraphy across East Antarctica with a
particular focus around the EPICA (European Project for Ice Coring in Antarctica) Dome C, Kohnen and Dome
F ice-core sites, Recovery Glacier (Humbert et al., 2018) and Dronning Maud Land (e.g., Steinhage et al., 2001;
Steinhage et al., 2013; Karlsson et al., 2018; Winter et al., 2019b; Wang et al., 2023). Additional significant
AWI RES surveys also span the Lambert and Recovery glacier catchments (Fig. 3f).
**3.6 Centre for the Remote Sensing of Ice Sheets / Operation IceBridge**
The USA-based University of Kansas began developing coherent RES systems in the 1980s but primarily
focussed on Greenland. A Kansas RES system with 150 MHz centre frequency was first deployed over
Antarctica in 2002 on a joint USA (NASA; National Aeronautics and Space Administration) / Chile (CECs;
Centro de Estudios Científicos) mission to survey fast-changing regions of West Antarctica (Rignot et al., 2004).
In 2005, Kansas became host to the USA's Center for Remote Sensing of Ice Sheets (CReSIS), an NSF-
designated national Science and Technology Centre with a focus on ice-sheet sounding[1]; and began to
operate an upgraded series of deep-looking RES systems named Multichannel Coherent Radar Depth
Sounders (MCoRDS). An early application of these RES systems was a wide-ranging survey of the Gamburtsev
Subglacial Mountains region of central East Antarctica in 2008/09 (Fig. 3g) that notably imaged multiple
basal-ice units disrupting the radiostratigraphy (Bell et al., 2011; and Sect. 5.4; Wolovick et al., 2014; Wrona
et al., 2018; and Sect. 5.4).
From 2009 to 2019, MCoRDS was frequently deployed onboard NASA's Operation IceBridge (OIB) programme,
which performed ten Antarctic RES campaigns collecting ~350,000 km of RES data (Fig. 3g; MacGregor et al.,
2021). Most surveys detected widespread radiostratigraphy using centre frequencies of ~190-194 MHz, but
for the 2009 to 2011 campaigns MCoRDS Version 1 the radiostratigraphic continuity is relatively poor
(MacGregor et al., 2021). From 2012, MCoRDS Versions 2 to 7 were introduced with progressively greater
power and bandwidth, significantly improving the detection of radiostratigraphy using frequencies in the
range of 150-450 MHz (Rodriguez-Morales et al., 2013; MacGregor et al., 2021). NASA OIB / CReSIS data have
been used to assess and track radiostratigraphy within the central East Antarctic Ice Sheet (Cavitte et al.,
2016; Winter et al., 2017), and across West Antarctica's central divide and Thwaites Glacier (Holschuh et al.,
2014; Koutnik et al., 2016; Bodart et al., 2021). Significantly, CReSIS pioneered early open access to processing
routines and radargrams (Liu et al., 2016), and continues to do so as part of the Open Polar Radar project
(Paden et al., 2021; and Sect. 6).

---

[1] From 2022 CReSIS, reflecting an expanding remit, was renamed the Center for Remote Sensing of *Integrated Systems*.



**3.7 Lamont-Doherty Earth Observatory**

From 2010, the Lamont-Doherty Earth Observatory (LDEO) of the USA's Columbia University developed an in-house Deep ICE Radar (DICE) RES system as part of an aerogeophysical suite ("IcePod") designed to be operated from LC-130 aircraft typically deployed by the US Antarctic Programme. DICE, with 188 MHz centre frequency and 60 MHz bandwidth, was operated between 2015 and 2017 to systematically survey the 500,000 km$^2$ Ross Ice Shelf (Fig. 3h; Tinto et al., 2019; Das et al., 2020).

**3.8 Polar Research Institute of China**

The Polar Research Institute of China (PRIC) has undertaken considerable airborne RES surveying across East Antarctica since 2015 (Fig. 3i). Deploying a 60 MHz centre-frequency RES system, which has heritage in the UTIG HiCARSII system (Sect. 3.4), configured in the "Snow Eagle 601" airborne platform, PRIC has systematically and extensively surveyed the Princess Elizabeth Land sector of East Antarctica (Cui et al., 2020b). Further profiling has also covered much of Mac. Robertson Land including Amery Ice Shelf (Cui et al., 2020a; Cui et al., 2020c). Several long profiles across Dome A, Ridge B, Vostok, Dome C and Wilkes Land (Cui et al., 2020a) could also be used to link radiostratigraphy with other RES campaigns across key sectors of East Antarctica. Recent efforts applying machine learning methods to the extraction of radiostratigraphy from these airborne RES data (Dong et al., 2021) show rich promise.

**3.9 Additional airborne RES datasets**

The RES providers discussed in the preceding sections have acquired >90% of the airborne RES data suitable for extracting internal architecture across the Antarctic ice sheets. Of the remainder (Fig. 3j), airborne RES data have been acquired, primarily with analogue systems, around parts of coastal East Antarctica by Antarctic programmes, institutions and universities from Australia (e.g., Morgan et al., 1982), Belgium (Van Autenboer and Decleir, 1975; using a SPRI RES system), Germany (by groups led from University of Münster, e.g., Thyssen and Grosfeld (1988) and the Federal Institute for Geosciences and Natural Resources (BGR), e.g., and the Federal Institute for Geosciences and Natural Resources (BGR), e.g., Damaske and McLean (2005)) and Italy (e.g., Frezzotti et al., 2004; Urbini et al., 2010). In West Antarctica, airborne RES data were acquired by the NSF in the 1970s across Ross Ice Shelf (Bentley, 1990) and in the 1980s across West Antarctica's Siple Coast region (Retzlaff et al., 1993) using a SPRI RES system; after which USA-led airborne RES surveys were arranged through the institutions already introduced above (Sect. 3.4 [UTIG], 3.6 [NASA/CReSIS] and 3.7 [LDEO]). More recently, the Korean Polar Research Institute has conducted airborne RES surveys around coastal East and West Antarctica with a system based on the UTIG MARFA RES system (e.g., Lindzey et al., 2020; Lee et al., 2021). Almost all of these campaigns, although we have broadly labelled them by national programmes or institutions, have relied on, and fundamentally been supported by, international collaboration in some or all of their component funding, logistical or scientific aspects. In most cases, because



they are now some decades old or not digitally rendered, the contemporary utility of these additional
airborne RES datasets for providing useful information on internal architecture remains largely to be
investigated, but some of them may yet prove instrumental in linking between two or more wider-ranging
surveys across parts of the ice sheet. The most promising, because they comprise several links between
coastal regions and the deep interior of East Antarctica at Dome C, were acquired by the Italian programme
under the auspices of EPICA (e.g., Tabacco et al. (1999) and Tabacco et al. (2008); plus see Siegert et al.
(2001b), for an example of a combined use of these data and those from the SPRI-NSF-DTU surveys of the
1970s).
**3.10 Ground-based RES datasets**
Since the 1960s, groups from at least twelve institutions have acquired ground-based RES datasets focussed
on sounding Antarctica's subglacial bed and have also typically imaged internal architecture in the process.
Typically, these ground-based surveys have been confined to smaller regions or shorter profiles than covered
by the airborne RES surveys, befitting the more common application of ground-based RES to detailed site
surveys in preparation for retrieving ice cores, or for accessing the ice bed or subglacial lakes (e.g., Frezzotti
et al., 2004; Laird et al., 2010; Christianson et al., 2012; Ross et al., 2020). From these surveys, several local
radiostratigraphies have been published (e.g., Eisen et al., 2005; Jacobel and Welch, 2005; Koutnik et al.,
2016; Cavitte et al., 2023; Chung et al., 2023; Koch et al., 2023). These detailed studies provide invaluable
seeding points for extending radiostratigraphies much more widely across the ice sheets (e.g., Winter et al.,
2019a) and for understanding better ice-sheet history and glaciological processes.
Supplementing the more local surveys, some ground-based profiles have been acquired over traverses of
multiple 100s of km over the Antarctic ice sheets, and these traverses, marked on Fig. 3k, merit special
attention as potential resources for analysing pan-continental radiostratigraphy. A particularly extensive
programme of ground-based surveys has been conducted since 1969 by the Japanese Antarctic Research
Expedition (JARE) connecting coastal East Antarctica in Dronning Maud and Enderby Land to Dome F, with
data from some of these traverses conducted in the 1990s underpinning seminal work on the origins of IRHs
(Fujita et al., 1999; Matsuoka et al., 2003). Today, data from JARE represent some of the most spatially
extensive of Antarctica's ground-based RES datasets and a rich repository of internal architecture (Fujita et
al., 2011; Van Liefferinge et al., 2021; Tsutaki et al., 2022). Further long ground-based RES traverses were
acquired by several national and international teams in the 2000s under the auspices of the International
Trans-Antarctic Scientific Expedition (ITASE). RES profiles containing particularly rich internal architecture
were acquired by the USA-NSF's ITASE traverses across both West (Welch and Jacobel, 2003; Jacobel and
Welch, 2005) and East Antarctica (Welch et al., 2009), with findings from Arcone et al. (2012a) suggesting
that in some parts of East Antarctica the radiostratigraphy is unconformable and may present significant
challenges to tracking radiostratigraphy.





Other institutes/consortia who have acquired wide-ranging and deep-looking ground-based RES profiles
extending 100s of km across the Antarctic ice sheets include the Australian National Antarctic Research
Expedition (ANARE; over Mac. Robertson and Princess Elizabeth Lands, traversing around ice feeding Amery
Ice Shelf - Craven et al. (2001); Wilkes Land - Jones and Hendy (1985); Medhurst (1985)); BAS (e.g., surveys
across West Antarctic catchments by King (2009); King (2011); Ross et al. (2011); Bingham et al. (2012);
Bingham et al. (2017); and Filchner-Ronne Ice Shelf; (Kingslake et al., 2016)); the Chilean Antarctic Institute
(Instituto Antártico Chileno, INACH, surveys around Institute Ice Stream including Subglacial Lake CECs; Rivera
et al. (2015); Napoleoni et al. (2020) - and connecting Institute Ice Stream to South Pole in a joint enterprise
with the Brazilian Antarctic Programme; Zamora et al. (2007)); the Russian Antarctic Expedition (traverses
connecting coastal stations to Vostok and Ridge B; (Popov, 2015; Popov, 2020)); PRIC (traverses connecting
coastal Zhongshan Station with Dome A; Luo et al. (2022)); and the International Thwaites Glacier
Collaboration (WAIS Divide to lower Thwaites Glacier between 2022 and 2024 using BAS and CReSIS ground-
based RES systems) (Fig. 3k). Especially for ice-core-related imaging of radiostratigraphy, deep-looking
ground-based surveys are still essential because of their high horizontal resolution.
**3.11 Summary**
Figure 3l collates the coverage of those RES datasets which were digitally acquired with GNSS navigation and,
in principle, represent the present coverage of existing RES data that could be used to develop a pan-Antarctic
radiostratigraphy. In practice, as the following section explores, only a small subset of these data have so far
been exploited, in part due to challenges in accessing data and working with them consistently, but mainly
because tracing and dating radiostratigraphy using existing methods is a highly time- and resource-intensive
process.
**4 Extracting and dating internal architecture from RES data**
The information available from radargrams (e.g. Fig. 2), and the degree to which the internal architecture can
be used for different applications, depend firstly on the settings of the RES system acquiring the data and
secondly on choices made in processing the data. Below we summarise the typical processing workflow for
radargram generation and highlight key decisions that influence interpretation of the resulting
radiostratigraphy. Figure 4 presents a conceptual support to this discussion. We then discuss the different
methods used to trace radiostratigraphy through radargrams, and to date key IRHs, and provide an inventory
of existing traced radiostratigraphy across Antarctica.
**4.1 Pulse compression, filtering, and image focussing for optimising IRH tracing**
RES can be categorised broadly based on two criteria: (a) Phase control of the transmitter or phase sampling
by the receiver (i.e., coherent vs. incoherent); and (b) the nature of the transmitted wave (pulsed versus

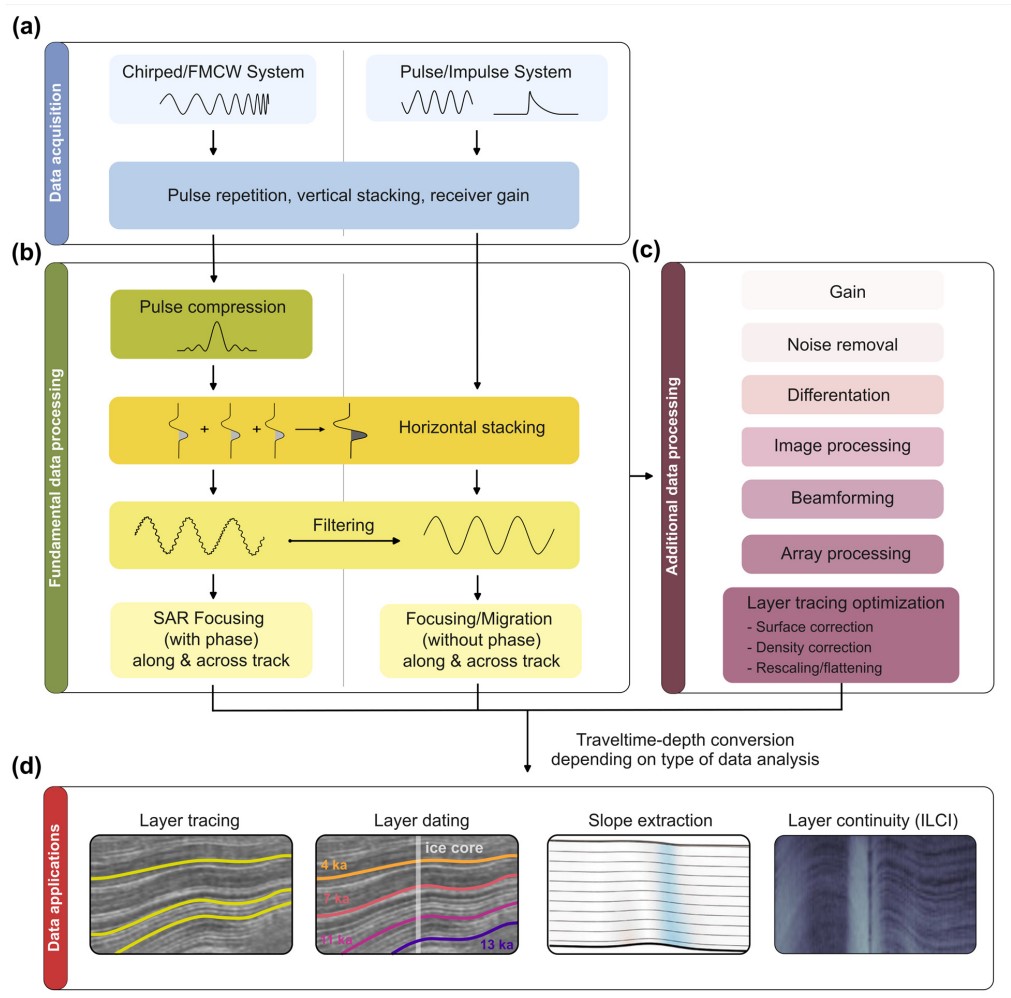

**Figure 4.** Flowchart illustrating key steps for the processing of RES data from chirp and pulse systems for subsequent radiostratigraphic analysis. (a) Basic configurations and parameters defined on data acquisition. (b) Fundamental and (c) additional steps commonly taken when processing data to visualise IRHs. (d) Depiction of some common ways of tracing or otherwise quantifying IRH geometry.


chirped; Gogineni et al. (1998); Peters et al. (2005)) (Fig. 4a). Processing is similar for all systems, so here we
highlight differences that affect radargram quality. Direct measurements of the dielectric properties of ice
cores show that ice conductivity varies on much smaller length scales than can be imaged by RES (Harrison,
1973; Eisen et al., 2003). Therefore, each RES system represents subsurface reflectors differently, and data
acquired from the same area but by different RES systems may show different IRHs on intersecting
radargrams due to the differences in RES imaging capabilities (see Fig. 5, after Winter et al. (2017), for an
example of a comparison between different RES systems). For pulsed systems, processing cannot improve



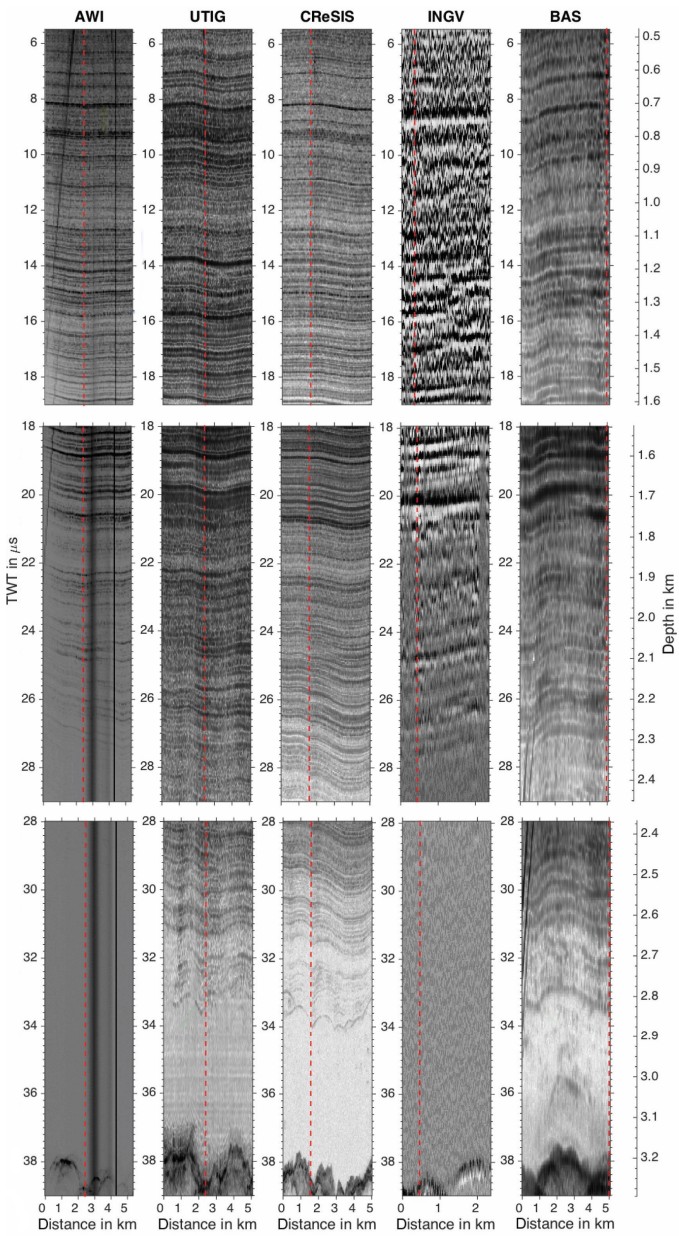

**Figure 5**. RES profiles of a few km length for five RES systems, that have profiled across or near EPICA Dome C. The vertical red line in each profile marks the position of the trace closest to Dome C. The surface reflections are shifted to time zero and the length of the RES profiles is indicated on the horizontal axes. For the bottom UTIG and CReSIS panels a 2-D-focused processing is applied. The RES data were acquired with: 1. AWI 150 MHz Aero-EMR; 2. UTIG 60 MHz HiCARS; 3. CReSIS 194 MHz MCoRDS; 4. Italian National Institute of Geophysics and Volcanology (INGV) 150 MHz RES system; and 5. BAS 150 MHz PASIN; for full details and original figure from which this is modified, see Winter et al. (2017).




the vertical resolution, which is controlled by the bandwidth and the rate of sampling of the received
waveform. For chirped systems, the waveform must be fully sampled first and then match-filtered,
integrating the received power while also finely resolving radiostratigraphy targets based on the chirp's
bandwidth (Hélière et al., 2007; Peters et al., 2007). This "pulse compression" is the first step in producing a
radargram from a chirped system.
Following initial data acquisition, RES data are typically processed using geophysical techniques of varying
sophistication (Fig. 4b). For example, incoherent noise is typically reduced by various forms of horizontal
averaging, and bandpass-filtering can remove irrelevant components of the measured signal. Finally, if
possible the data should be focused or migrated to reposition the received signal energy as precisely as
possible to their true subsurface locations. This can be done via several methods: (a) Incoherent echo
summation, often termed *migration* as in reflection seismology (Yilmaz, 2001); (b) SAR-focusing for point
scatterers, common in satellite applications (Ulaby and Lang, 2015); or (c) algorithms designed specifically
for RES of specular reflections (Heister and Scheiber, 2018; Castelletti et al., 2019; Xu et al., 2022). SAR-
focusing has a proven ability to reduce image artefacts and improve along-track resolution, especially in areas
with steeply-sloping radiostratigraphy (Holschuh et al., 2014; Castelletti et al., 2019). Multiple SAR-processing
techniques currently exist for coherent RES systems, including: (a) unfocused SAR (short apertures without
phase correction and equivalent in name to Doppler filtering or coherent echo summation; Hélière et al.
(2007)); or (b) more advanced focused SAR, using either 1-D correlations resulting in intermediate apertures,
or 2-D correlations resulting in longer apertures (Peters et al., 2005; Peters et al., 2007). The latter is the
processing of choice for modern coherent systems for the detection of IRHs in areas with steeply dipping
reflections. Unfocused and 1-D SAR approaches will emphasise flat specular reflectors and reduce clutter, at
a cost of dipping specular horizons. Large SAR apertures are critical for tracking steeply dipping IRHs, but
present greater computational costs and an overall reduction of signal to noise ratio. Cross-track antenna
arrays can allow for determination of cross-track IRH slopes.
A series of additional corrections and image-processing steps can also be taken to optimise RES data for
tracing radiostratigraphy (Fig. 4c). For radar data acquired by airborne platforms, the aircraft-to-ice surface
space on the radargram must be removed to obtain true depths below the ice surface; this is often conducted
by shifting the vertical axis of the radargram to time zero for each RES trace and flattening the surface based
on the location of the surface reflection on the radargram. This can be done using data from the altimeter
and/or LIDAR onboard the aircraft, high-resolution surface DEMs, or using the picked surface reflection from
the radargram itself (e.g., MacGregor et al., 2015a). Localised density corrections, based on ground-truthing
measurements in the upper section of ice cores or other geophysical measurements (e.g., radar data acquired
by airborne platforms; Eisen et al., 2002), may also be applied to convert the two-way-travel time from the
RES data to ice-equivalent depths. Alternatively, for depth-correcting RES below the pore close-off depth, a
spatially uniform density value that is typically of the order of several metres may be used to obtain ice-





equivalent depths (e.g., Ashmore et al., 2020), although this assumption may only be valid in dry and stable
parts of the ice sheet and not in highly dynamic regions (Dowdeswell and Evans, 2004). Others have also
vertically rescaled (or flattened) RES data to facilitate the tracing of continuous reflections by semi-automatic
pickers (e.g., Fahnestock et al., 2001a; Sect. 4.2; MacGregor et al., 2015a). Finally, specific image-processing
filters can also be applied to enhance the gain and reduce incoherent noise, which can facilitate IRH tracing
on RES data (Ashmore et al., 2020; Bodart et al., 2021; Wang et al., 2023).
Importantly for users interested in tracing IRHs, and especially the deepest IRHs, most RES data over
Antarctica, including those available from open-access repositories, are not optimised for detecting
radiostratigraphy. Typically the data have been acquired and processed to optimise retrieval of the bed echo,
and some datasets require considerable reprocessing from the raw data to improve the clarity of the
radiostratigraphy between the ice surface and the bed (Castelletti et al., 2019). In particular, for thick or
unusually heterogenous ice, the best strategy is often to experiment with filtering data differently at different
depths until the IRHs at selected depths are most clearly visualised.
**4.2 Tracing radiostratigraphy**
The primary method for extracting internal architecture from radargrams has been to trace or "pick" IRHs,
typically using semi-automated techniques (e.g., Cavitte et al., 2016; Koch et al., 2023). Where radargram
quality is high, IRHs are easily traced and continuous, and automated methods may also perform well (e.g.,
Panton, 2014; Xiong et al., 2018; Delf et al., 2020). Machine-learning methods show promise for more rapidly
tracing radiostratigraphy in new datasets; but so far successful applications have been limited to shallow IRHs
in the upper few tens of metres of the ice column (e.g., Dong et al., 2021; Rahnemoonfar et al., 2021; Yari et
al., 2021). Thus, for most radargrams and deep-ice applications, semi-automated tracing of IRHs is presently
required. This relies on algorithms that typically follow the local maxima in return power between adjacent
traces within a predetermined vertical window, using either open-source or commercial and bespoke
software from the seismic industry (e.g., Winter et al., 2019a; Ashmore et al., 2020; Sanderson et al., 2024).
A comprehensive overview of IRH-tracing methods is provided by Moqadam and Eisen (2024).
The process of tracing IRHs can be categorised into two main approaches: (a) tracing as many IRHs as possible
regardless of their amplitudes or continuity (MacGregor et al., 2015a); or (b; more commonly) by identifying
IRHs that have a high echo-power, appear distinguishably brighter than adjacent IRHs on radargrams and are
continuous for long distances (>100 km), using crossovers between intersecting RES profiles to ensure
reliability in the tracing process (e.g., Cavitte et al., 2016; Winter et al., 2019a; Ashmore et al., 2020; Bodart
et al., 2021; Wang et al., 2023).
Importantly, the thickness of a given IRH in a radargram is dependent on the range resolution of the RES
system used to image it, such that RES systems with high pulse-width, and thus finer vertical resolution, may



detect several thinner IRHs that would otherwise appear as a single, broader reflection in coarser-resolution
systems (see Fig. 5 and Harrison, 1973; Millar, 1982; Karlsson et al., 2014; Winter et al., 2017; Bodart et al.,
2021; Cavitte et al., 2021). This must be accounted for when comparing the position and aspect of IRHs traced
in data from RES systems operating with different frequencies and system characteristics (Winter et al., 2017).
**4.3 Complementary approaches to tracing IRHs for characterising radiostratigraphy**
Even having applied all possible data processing strategies described above, radiostratigraphy may remain
challenging or impossible to trace over some regions due to the innate physical properties of ice in such areas.
For example, IRHs may become warped/buckled or disrupted by differential ice flow or flow over steep
topography (e.g., Siegert et al., 2003b; Ross et al., 2011; Bingham et al., 2015; Franke et al., 2023; Jansen et
al., 2024), while unconformities can be introduced by significant wind scouring of the ice surface (e.g., Welch
and Jacobel, 2005; Luo et al., 2022). This variability in itself provides important information about past and
present ice behaviour (as we explore further in Sect. 5), and hence warrants alternate methods to
characterise the radiostratigraphy where IRHs cannot readily be traced.
One method for assessing the general variability of radiostratigraphy across large regions of ice sheets is the
Internal Layering Continuity Index (ILCI) developed by Karlsson et al. (2012). This tool maps the variability in
vertical signal strength for individual RES traces, acting as a relative measure of the number of dielectric
contrasts compared to signal-to-noise ratio. High ILCI values typically indicate regions of an ice sheet
characterised by multiple, traceable IRHs, while low ILCI values tend to indicate regions of ice sheet with
disrupted or discontinuous IRHs or regions with very few or no IRHs detected by the RES system. Although
the method is not easily transferable between different RES systems due to acquisition and processing
differences, ILCI has been extensively applied to several regions both in Antarctica (Fig. 6) and Greenland as
a mechanism for identifying rapidly the specific sub-regions in which IRHs are likely to be traceable (e.g., Sime
et al., 2014; Bingham et al., 2015; Karlsson et al., 2018; Frémand et al., 2022; Tang et al., 2022; Sanderson et
al., 2023).
Alternative methods have focussed on the extraction of IRH slopes. This avenue acknowledges the challenges
of tracing and dating radiostratigraphy in areas of fast or complex ice flow, or where the acquisition or
processing methods that have been used were not tailored to the recovery of radiostratigraphy. For
discontinuous radiostratigraphy, local slope information is valuable, because radiostratigraphic slope is
closely related to particle trajectories within the ice sheet (Hindmarsh et al., 2006; Parrenin and Hindmarsh,
2007; Ng and King, 2011; Holschuh et al., 2017). Several methods have therefore been developed to extract
slope information, such as incoherent averaging methods (Sime et al., 2011; Holschuh et al., 2017; Delf et al.,
2020) and methods that use along-track phase information during SAR processing to estimate IRH slope
(MacGregor et al., 2015a; Castelletti et al., 2019; Oraschewski et al., 2023).



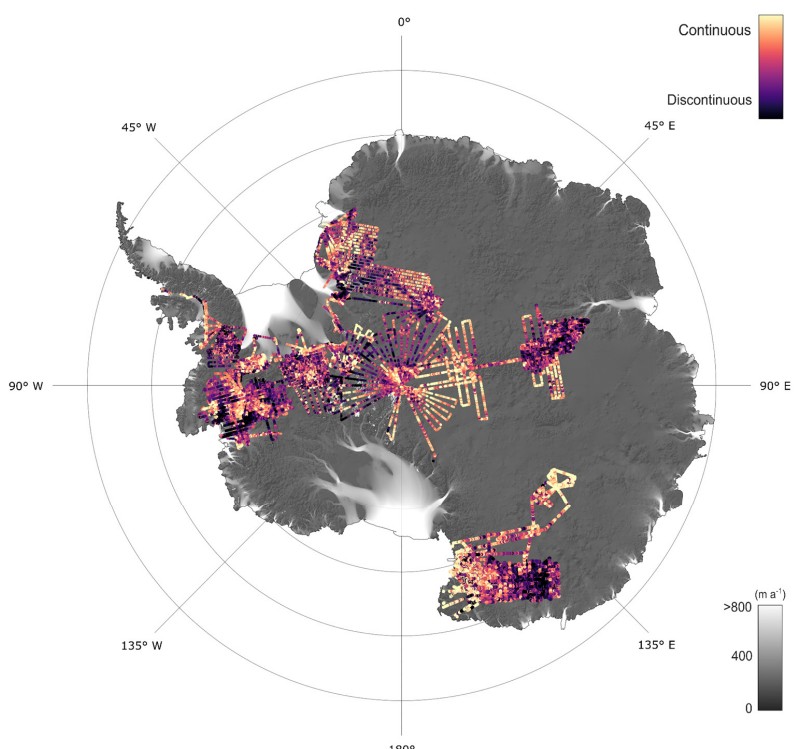

**Figure 6.** Radiostratigraphic continuity (ILCI) calculated over 10 airborne RES datasets acquired by BAS. Continuous and readily traceable IRHs are indicated in the slow-flowing regions of the ice sheet (high ILCI; bright yellow) whereas disrupted or absent IRHs are likely in the faster-flowing sections of ice streams or where subglacial topography is highly variable (low ILCI; dark purple). The background maps show ice-flow velocities from MEaSUREs (Rignot et al., 2017) and a hillshade of the bedrock from BedMachine (Morlighem, 2020). Figure modified from Frémand et al. (2022).


**4.4 Dating internal-reflection horizons (isochrones)**

As introduced in Sect. 2, most RES-imaged IRHs have been shown to be isochronous, and the majority of
those we treat in this review (i.e. that are imaged in between the first and last few hundreds of metres of the
ice column) arise due to the RES systems imaging variations in the electrical conductivity (i.e. acidic content)
of the ice with depth. Hereafter in this paper, reiterating that most IRHs are isochrones, we will use the term
isochrones to refer to IRHs, and will only re-use the term IRH where it may be ambiguous concerning whether
IRHs are isochronous.
Ages can be assigned to isochrones at intersections with deep ice cores where age-depth models have already
been derived from chemistry analyses (e.g., McConnell et al., 2017; Cole-Dai et al., 2021; Bouchet et al., 2023),
but also using modelling techniques where this is not possible. Before any age can be assigned, the age



uncertainty that arises from the RES system itself must first be assessed. Uncertainty in reflector depth arises
from several sources: (a) proximity of the RES profile to the ice-core site, otherwise a specific reflector
geometry (typically flat) must be assumed between the point of closest approach and the ice-core site
(MacGregor et al., 2015a); (b) the radio-wave speed, which varies based on permittivity variations as a
function of englacial density and anisotropy (e.g., Kovacs et al., 1995; Fujita et al., 2000); (c) the range
resolution of the RES system and the signal-to-noise ratio of each traced reflection at (or near) the ice-core
site, which enable an estimate of the depth precision to which each traced reflection can be known (e.g.,
Cavitte et al., 2016); and (d) the picking accuracy of both the ice surface and the isochrones themselves,
which can add several metres of uncertainty. This latter point may include the uncertainty arising from the
source of the surface product (i.e. either from cm-resolution onboard altimeter/LIDAR), or directly from the
RES data which have much lower resolution of the order of several metres); and whether the picking
algorithm is tailored to extract the onset of the reflection, the half-amplitude, or the peak value.
The ideal scenario for assigning ages to isochrones is that a RES profile intersects or passes sufficiently close
(~500 m vicinity) to the location of an ice-core site for the ice core's depth-age scale (from chemical profiling
or layer counting) to be useable for directly assigning ages to the RES-imaged isochrones. In such cases, the
isochrone-depth uncertainty can then be combined with the ice-core age uncertainty to assign a total age
uncertainty to the mapped reflections; in these cases, uncertainty is generally dominated by the ice-core-
derived age uncertainty in the upper third of the ice column, while the RES-derived depth uncertainty
increasingly dominates at larger depth (e.g., MacGregor et al., 2015a; Cavitte et al., 2016; Muldoon et al.,
2018; Winter et al., 2019a; Wang et al., 2023). More recently, some isochrones have been dated not by their
direct intersection with an ice core, but rather by intersecting other RES datasets that in turn have already
been dated by their intersection with a distant ice core. In these cases, the age-depth profile is transferred
to the new dataset at the crossover(s) between the intersecting RES datasets (e.g., Ashmore et al., 2020;
Bodart et al., 2021). In these cases, the relative uncertainties of the different RES systems at the intersections
between RES datasets additionally need be factored into the final age estimation, and the final age estimates
are commonly checked using the modelling techniques introduced next (e.g., Bodart et al., 2021; Sanderson
et al., 2024).
Where isochrones cannot be directly correlated to an ice-core age-depth relationship due to a lack of nearby
ice cores, any intersections with previously dated isochrones, or missing sections in the record (e.g., due to
disrupted englacial stratigraphy), age-depth modelling is required to assign ages to isochrones. This is
typically done using 1-D models in stable parts of the ice sheet such as at ice divides (e.g., Nye, 1957;
Dansgaard and Johnsen, 1969; Ashmore et al., 2020; Bodart et al., 2021; Sanderson et al., 2024); or using
more complex multidimensional (2D/3D) models in areas with challenging ice-flow or bed conditions (e.g.,
Waddington et al., 2007; MacGregor et al., 2015a; Parrenin et al., 2017; Lilien et al., 2021).




### 4.5 Existing dated radiostratigraphy across Antarctica

Before the inception of *AntArchitecture* in 2018, several studies had produced radiostratigraphies spanning the last 17.5 ka across West Antarctica and 352 ka for East Antarctica (e.g., Hodgkins et al., 2000; Siegert and Hodgkins, 2000; Siegert, 2003; Siegert and Payne, 2004; Jacobel and Welch, 2005; Leysinger Vieli et al., 2011; Steinhage et al., 2013; Karlsson et al., 2014; Wang et al., 2016). However, the spatial extents of these radiostratigraphies were relatively limited. Through *AntArchitecture*, a more coordinated and focused approach to characterising Antarctic radiostratigraphy has been conducted, as depicted in Figure 7 and detailed in Table 1. This programme has facilitated the recovery and characterisation of several isochrones with ages up to 25 ka across much of the Amundsen and Weddell Sea sectors of West Antarctica (Muldoon et al., 2018; Ashmore et al., 2020; Bodart et al., 2021; Bodart et al., 2023). Over East Antarctica, a much older record has been extracted, owing to the more stable and slow-flowing ice conditions in the area, including isochrones dating back to the last 705 ka (Cavitte et al., 2016; Winter et al., 2019a; Beem et al., 2021; Cavitte et al., 2021; Chung et al., 2023; Wang et al., 2023; Sanderson et al., 2024).

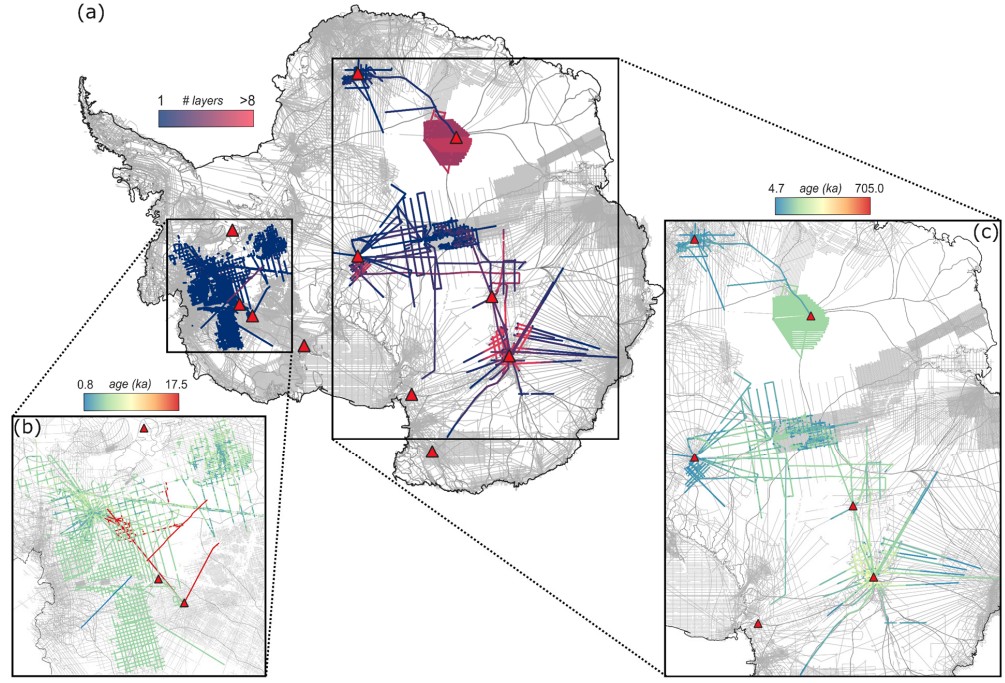

**Figure 7.** Existing open-access dated stratigraphies across Antarctica obtained from the Digital Object Identifiers (DOIs) provided in Table 1, with RES profiles for Bedmap-2 and Bedmap-3 products shown in the background (grey; Frémand et al., 2023). Existing deep ice cores (defined here as ice cores that have been drilled to near the ice-bed interface and that provide a multi-millennial record) are shown as red triangles. (a) Maximum number of layers traced through each dataset (from 1 to >8). (b-c) age of the deepest (oldest) layer across each dataset for the WAIS (b) and EAIS (c) regions respectively.






**Table 1.** Inventory of expansive radiostratigraphic datasets for the Antarctic ice sheets, ordered by region
and length (km of RES profiles) of dataset. The data are mapped in Figure 7; locations of ice cores are marked
on Fig. 3a. DOIs are provided where the underlying isochronal data are available in open-access format. Data-
provider acronyms are expanded in Sect. 3 of the text; in most cases we also list here a specific project
acronym for each survey which can be cross-referenced through the reference and/or dataset listed in each
row.
(For EGUsphere formatting, this 10-column table is presented across two pages.)

| Region | Survey dates | Data provider (cf. Sect. 3) | Survey name / acronym | Ice-core intersection(s) | No. of traced isochrones |
|---|---|---|---|---|---|
| EAIS | 1998 - 2008 | AWI / CReSIS | DoCo / EPICA / AGAP | Kohnen / Vostok / Dome C | 5 |
| EAIS | 2016 - 2017 | AWI | Beyond EPICA Dome Fuji | Kohnen / Dome F | 7 |
| EAIS | 2008 - 2018 | UTIG | ICECAP | Dome C | 26 |
| EAIS | 2007 - 2016 | BAS | AGAP / PolarGap | South Pole | 3 |
| EAIS | 1974 – 1979 | SPRI-NSF-DTU | - | Vostok / Dome C | 12 |
| EAIS | 1974 – 1979 | SPRI-NSF-DTU | - | Vostok / Dome C | >32 |
| EAIS | 2016 – 2017 | PRIC | South Pole Corridor | South Pole | 8 |
| EAIS | 2016 – 2018 | BAS | Beyond EPICA Little Dome C | Dome C | 20 |
| EAIS | 2002 – 2003 | AWI | - | Dome F | 8 |
| EAIS | 1974 – 1979 | SPRI-NSF-DTU | - | Vostok | 15 |
| EAIS | 2004 – 2005 | PRIC | Dome A (CHINARE-21) | Vostok | 6 |
| WAIS | 1991 – 2014 | UTIG | CASERTZ / SOAR / AGASEA / GIMBLE | Byrd / WAIS Divide | 1 |
| WAIS | 2004 – 2018 | BAS / CReSIS | BBAS / OIB | WAIS Divide | 4 |
| WAIS | 2010 – 2011 | BAS | IMAFI | - | 3 |
| WAIS | 2000 – 2001 | NSF | ITASE | Byrd | 1 |
| WAIS | 1977 – 1978 | SPRI-NSF-DTU | - | Byrd | 5 |




**Table 1** continued: Columns 6-10.

| Isochrone age range (ka) | Length of traced IRHs (km) | Reference | Dataset DOI |
|---|---|---|---|
| 38.0 – 161.0 | 40 000 | Winter et al. (2019a) | 10.1594/PANGAEA.895528 |
| 31.4 – 232.7 | 20 000 | Wang et al. (2023) | 10.1594/PANGAEA.958462 |
| 10.0 – 705.0 | 15 500 | Cavitte et al. (2021) | 10.15784/601411 |
| 38.0 – 162.0 | 13 000 | Sanderson et al. (2024) | 10.5285/cfafb639-991a-422f-9caa-7793c195d316 |
| 17.5 – 352.4 | 8 000 | Leysinger Vieli et al. (2011) | 10.1029/2010JF001785 |
| 45.9 – 169.7 | 4 000 | Siegert (2003) | - |
| 4.7 – 93.9 | 2 000 | Beem et al. (2021) | 10.15784/601437 |
| 10.5 – 414.6 | 1 280 | Chung et al. (2023) | 10.1594/PANGAEA.963470 |
| 4.7 – 72.4 | 1 200 | Steinhage et al. (2013) | - |
| 17.0 – 211.0 | 1 000 | Leysinger Vieli et al. (2004) | - |
| 34.3 – 161.4 | 215 | Wang et al. (2016) | - |
| 4.7 | 19 000 | Muldoon et al. (2018) | 10.15784/601673 |
| 2.3 – 16.5 | 15 000 | Bodart et al. (2021) | 10.5285/f2de31af-9f83-44f8-9584-f0190a2cc3eb |
| 1.9 – 8.1 | 6 000 | Ashmore et al. (2020) | 10.5281/zenodo.4945301 |
| 17.5 | 1 850 | Jacobel and Welch (2005) | 10.7265/N5R20Z9T |
| 0.8 – 16.0 | 800 | Siegert and Payne (2004) | 10.1002/esp.1238 |






A notable finding is the presence of widespread and ubiquitous isochrones that have been imaged by
different RES systems and are found in several ice-core records. Across West Antarctica, the most ubiquitous
isochrone, dated precisely and independently at Byrd and WAIS Divide ice cores to ~4.7 ka, has been
identified by several studies (Jacobel and Welch, 2005; Karlsson et al., 2014; Holschuh et al., 2018; Muldoon
et al., 2018; Ashmore et al., 2020; Table 1; Bodart et al., 2021; Bodart et al., 2023). There is evidence that this
same isochrone may also be found widely across East Antarctica, based on sulphate concentrations in ice
cores and findings from individual RES surveys across the region (Steinhage et al., 2013; Winski et al., 2019;
Beem et al., 2021; Cole-Dai et al., 2021; Sigl et al., 2022). Additionally, across much of the West Antarctic Ice
Sheet an isochrone dated at 17.5 ka has been observed in both ground-based and airborne RES data (Jacobel
and Welch, 2005; Muldoon et al., 2018; Bodart et al., 2021; Table 1). This 17.5 ka RES isochrone has been
identified and linked to an eruption from West Antarctica's Mount Takahe in both the Byrd (Hammer et al.,
1997) and WAIS Divide (McConnell et al., 2017) ice cores. Over East Antarctica, packages of closely spaced
isochrones of ages ~38 ka, ~73 ka, ~128 ka, ~160 ka, and ~170 ka have been traced from ice cores (Leysinger
Vieli et al., 2011; Winter et al., 2019a; Cavitte et al., 2021; Table 1; Wang et al., 2023; Sanderson et al., 2024);
notably, the ~73 ka isochrone has been linked by ice-core profiling to the Toba Eruption in Indonesia
(Svensson et al., 2013). Together, such distinct isochrones, imaged by and from multiple RES systems and
platforms, provide important regional or continental-wide time markers, equivalent to Greenland's highly
recognisable "three sisters" (Fahnestock et al., 2001a; MacGregor et al., 2015a) for inferring past changes at
specific time intervals.
Despite the advances discussed here, the established radiostratigraphy across the Antarctic ice sheets
currently represents only a small subset of the total available RES data (Fig. 7, and refer back to Sect. 3 and
Fig. 3). The establishment of the *AntArchitecture* community, and its commitment to establish protocols for
sharing and processing internal architecture across the multiple datasets, is expected to facilitate further
isochrone tracing, which will in turn contribute to the development of the first three-dimensional age-depth
model of the ice sheet.
**5 Applications of internal architecture to wider Antarctic science**
Here, we now review to what scientific purposes internal architecture has already been exploited. Sect. 5.1
to 5.4, supported by Figure 8, exemplify four primary applications of RES-imaged isochrones, Sect. 5.5
explores the scientific applications of other forms of internal architecture, and Sect. 5.6 discusses how
radiostratigraphic data have been incorporated into numerical modelling, and their use in calibrating ice-
sheet models of varying complexity. This section contextualises the following Sect. 6 which then suggests
priorities for future research that will be enabled as Antarctica's internal architecture, and particularly its
radiostratigraphy, continue to be explored and made available.





**Figure 8.** Schematic illustration of radiostratigraphic observations within an ice sheet and their scientific applications; (a), in the centre, depicts typical ice-sheet locations for applications shown in subsequent panels. (b) Connecting and validating ice cores in Greenland (after MacGregor et al., 2015a). (c) Imaging intersections of IRHs with ice surface in region of surface wind scouring (after Winter et al., 2016). (d) Using isochrones to calculate basal melting across Subglacial Lake Vostok (after Siegert et al., 2001a). (e) Using isochrone drawdown to locate region of elevated geothermal heat flux near South Pole (after Jordan et al., 2018). (f) Application of "Internal Layering Continuity Index" (ILCI) to quantify disruption (folding/warping) to otherwise continuous isochrones (after Bingham et al., 2015). (g) Using intersecting RES profiles to explore ice anisotropy (after Gerber et al., 2023). (h) Raymond Arch imaged in shallow (top panel) and deep RES across Derwael Ice Rise, Dronning Maud Land (after Drews et al., 2015). (i) Basal-ice units and suggested accreted basal ice in East Antarctica (after Bell et al., 2011). (j) Basal crevasses imaged in West Antarctica and used to date regrounding of previously floating ice (after Kingslake et al., 2018). (k) Prominent tephra horizon imaged by RES across Pine Island Glacier, West Antarctica (after Corr and Vaughan, 2008).




### 5.1 Radiostratigraphy and ice cores

Ice cores from Antarctica provide fundamental palaeoclimate records (e.g., EPICA Community Members, 2004; WAIS Divide Project Members, 2015), and we have already introduced the concept that RES records tied to existing ice cores provide a basis for extending these "point-source" age-depth chronologies into 3-D age-depth fields that extend widely across the Antarctic ice sheets (Sect. 4; especially 4.4 and 4.5). Conversely, RES-imaged radiostratigraphy can be used to guide the best locations for recovering future ice cores. Accumulation rate, ice dynamics and age-depth relationships extracted from isochrones have previously informed the appropriateness of coring sites (e.g., Neumann et al., 2008; Parrenin et al., 2017; Beem et al., 2021; Wang et al., 2023) and have been essential for pre-site survey of potential future ice coring, e.g. for the *Oldest Ice* endeavour of the International Partnerships for Ice Core Sciences (IPICS; e.g., Fischer et al., 2013; Van Liefferinge and Pattyn, 2013; Karlsson et al., 2018; Lilien et al., 2021; Chung et al., 2023).

Radiostratigraphy has also provided opportunities for synchronising and reducing uncertainties in ice-core chronologies by facilitating the direct tracing of isochrones between two or more ice cores in order to correlate ice-core chronologies (as achieved for the Greenland Ice Sheet by MacGregor et al., 2015a; see Fig. 8b). In Antarctica, previous studies that have used isochrones to correlate chronologies between ice cores include Siegert et al. (1998), Steinhage et al. (2013), Cavitte et al. (2016) Le Meur et al. (2018) and Winter et al. (2019a) for East Antarctica, and Muldoon et al. (2018) for West Antarctica. These studies have provided confidence that ice cores obtained from locations separate by 100s of km capture analogous variations in palaeoclimate at regional scales, and that the signals recorded by RES correspond to genuine physical variations in the ice (typically variations in electrical conductivity, often related to fallout from past volcanic eruptions; as noted in Sect. 4.5).

The key challenge in synchronising ice-core records between distant sites using RES has been in resolving the radiostratigraphically- and ice-core-derived chronologies between each ice-core site, given the order-of-magnitude difference in resolution of chronologies recoverable from RES (on the order of metres) versus ice-core records (on the order of centimetres). This has typically been dealt with using forward modelling based on electrical-conductivity measurements or dielectric profiling of the ice cores to provide a transfer function (e.g., Miners et al., 1997; Hempel et al., 2000; Eisen et al., 2003; Eisen et al., 2006; Winter et al., 2017; Mojtabavi et al., 2022), or by adopting Bayesian frameworks which provide a probability distribution of the age of the isochrones (Muldoon et al., 2018). Thus, while the age-depth fields compiled from isochrones will never match the precision and accuracy of ice-core age-depth relationships (MacGregor et al., 2015a; Winter et al., 2017), they provide the spatial context that 'point-source' ice cores cannot. Through isochrone-constraint modelling (see Sect. 5.6), the age of the ice and its spatial distribution can be more effectively constrained in regions distant from the current drilling sites (Born and Robinson, 2021; Sutter et al., 2021).





In marginal locations of the ice sheets, or around nunataks, where persistent pronounced surface scouring is
co-located with upward ice flow over subglacial topography – i.e., in regions of so-called "blue ice" – very old
ice may outcrop obliquely to the ice surface and hence allow the recovery of a "horizontal ice core" along
the ice surface (Spaulding et al., 2013). Dated isochrones have been used to trace the age-depth model
recovered from horizontal ice cores back into the ice sheet (Reeh et al., 2002; Siegert et al., 2003a; Winter et
al., 2016; Fogwill et al., 2017; Baggenstos et al., 2018; see Fig. 8c). However, shearing and folding can disrupt
the stratigraphic order of the outcropping IRHs, rendering the interpretation of their radiostratigraphy more
complex than for most vertical ice cores.
**5.2 Surface mass balance**
In slow-flowing ice and especially around ice divides, the depth of isochrones is largely controlled by surface
mass balance and therefore dated radiostratigraphy has made it possible to reconstruct past surface mass
balance over millennial timescales across spatially extensive regions (e.g., Nereson et al., 2000; Siegert, 2003;
Siegert and Payne, 2004; Eisen et al., 2005; Waddington et al., 2007; Neumann et al., 2008; MacGregor et al.,
2009; Leysinger Vieli et al., 2011; Karlsson et al., 2014; Koutnik et al., 2016; Cavitte et al., 2018; Bodart et al.,
2023). Such records have fundamentally informed us about how mass balance has changed with time over
past millenia, for example showing that accumulation rates changed significantly over central (Siegert and
Payne, 2004; Neumann et al., 2008; Koutnik et al., 2016; Bodart et al., 2023) and coastal (Karlsson et al., 2014)
West Antarctica throughout the Holocene. Typically, vertical strain rates must be corrected for the whole ice
column, particularly in regions of (present or past) fast flow, or there is a need to account for basal processes
such as enhanced basal melting (e.g., Leysinger Vieli et al., 2011; Chung et al., 2023), because in such cases
the isochrone depths will be dynamically modified and therefore will not represent the surface mass balance
at the time of deposition (e.g., Koutnik et al., 2016). Where the radiostratigraphy has not been impacted
significantly by strain, the shallow-layer approximation can be applied, which allows us to ignore these strain-
rate corrections (Waddington et al., 2007). If horizontal advection influences the stratigraphy 2D, 2.5D or 3-
D modelling is required (see Sect. 5.6).
Regions of unconformable radiostratigraphy occurring throughout the ice column in parts of Antarctica have
partly limited the extent to which some surface mass balance records could be more widely extrapolated
(Arcone et al., 2012b; Cavitte et al., 2016). RES surveys of the upper ~100 m of the ice column in the affected
regions typically reveal widespread conformal, annual horizons modified by local variations in accumulation
or ice flow (Eisen et al., 2008), and the majority of them have been ascribed to wind scouring out surface
deposits and forming "megadunes" (Das et al., 2013; Traversa et al., 2023) that then become progressively
buried as sets of unconformable IRHs. Studies have identified such unconformities in several locations in East
Antarctica (Welch and Jacobel, 2005; Traversa et al., 2023) and West Antarctica (Woodward and King, 2009;
Holschuh et al., 2018).





### 5.3 Basal melting and geothermal heat flux

Isochrones have been used to calculate melting at the base of the ice exploiting the principle that melting from the presence of a subglacial water body or enhanced geothermal heat flux draws isochrones down towards the ice base. Mismatches between surface-accumulation-driven modelled isochrones and traced isochrones have been used to infer regions of enhanced basal melting in Greenland (Dahl-Jensen et al., 1997; Fahnestock et al., 2001b) and Antarctica (Carter et al., 2009) on the principle that removal of ice at the base by basal melting thins annual layers above. However, for locating areas of enhanced geothermal heat flux (or subglacial lakes, which may sometimes owe their existence to enhanced geothermal heat flux) researchers now typically rely more on analysing the reflectivity or specularity of the ice-bed echo in RES data (e.g., Young et al., 2016; Chu et al., 2021), and only use isochrones to guide derivations of basal melting where such more direct data are lacking.

Isochrones have been analysed in more detail over parts of Antarctica to constrain basal melting in more localised settings. For example, Siegert et al. (2000) used deviations in the dip of deep isochrones away from parallelism with the ice-bed/subglacial-lake surface over Subglacial Lake Vostok to calculate basal melting and water exchange between the lake and the overlying ice sheet (Fig. 8d). Jordan et al. (2018) identified isochrones dipping towards the bed ~200 km from the South Pole (Fig. 8e), and used these to model how much basal melt would be required to draw the isochrones down towards the bed. By assuming that minimal frictional melting would be generated by the slow ice flow in this region, they showed that the most likely cause of the isochrones being drawn down towards the bed must be enhanced geothermal heat flux in this region. Ross and Siegert (2020) undertook a detailed survey of isochrone geometry over Subglacial Lake Ellsworth, West Antarctica, and showed that the isochrones were preferentially drawn down over the NW shoreline of the lake, rather than the lake itself. This conclusion was in agreement with the pattern of basal mass balance derived from previous numerical modelling of water circulation in the lake and indicated very high basal melting of ~16 cm a$^{-1}$ on its northern shoreline.

### 5.4 Ice-flow dynamics

Present-day (last ~35 years) information on ice-flow dynamics is derived from satellite monitoring of ice-surface flow (Rignot et al., 2017), but to understand fully where and how ice-flow dynamics have changed over the past several thousand years, and hence may be likely to do so again, researchers have interrogated how changes to ice-flow dynamics have been imprinted into the RES-imaged internal architecture. The most common methodology has been to explore and classify where the radiostratigraphy diverges from relatively flat isochrones to profiles that show folding (a.k.a. buckling, warping or disruption) of the isochrones (Fig. 8f). Wherever there is folding of isochrones, and we assume they were originally deposited as flat layers, it is an indication that the ice has experienced considerable strain, often as a result of flowing around or over significant bedrock obstacles (Robin and Millar, 1982; Hindmarsh et al., 2006; Tang et al., 2022) or becoming



variously stretched and compressed as it flows through an ice-stream onset region or through ice-stream
shear margins (Jacobel et al., 1993; Bell et al., 1998; Ng and Conway, 2004; King, 2011). Overall, isochrone
folding can indicate convergent ice flow, anisotropic rheology, basal freeze-on, basal sliding, non-negligible
transverse velocity gradients, or the abutting of units of contrasting rheology. Importantly, the signature
recorded by these processes is often advected downstream, so that where it is observed does not necessarily
indicate where the folding took place (Weertman, 1976; Jacobel et al., 1993; Leysinger Vieli et al., 2004;
NEEM Community Members, 2013; Wolovick et al., 2014; Bons et al., 2016; Leysinger Vieli et al., 2018; Ross
et al., 2020; Franke et al., 2021; Jennings and Hambrey, 2021; Jansen et al., 2024). In certain cases, relict folds
that do not correspond to the current ice-flow direction indicate a past change in ice-flow direction (Conway
et al., 2002; Siegert et al., 2004; Rippin et al., 2006; Franke et al., 2022).
While, therefore, there are multiple origins for isochrone folding, their geographical association with fast ice
flow has led to their presence being used as a broad diagnostic of the long-term stability (or otherwise) of ice
flow around Antarctica (e.g., Rippin et al., 2003b; Siegert et al., 2003b; Bingham et al., 2007; Karlsson et al.,
2009; Ross et al., 2011; Bingham et al., 2015; Winter et al., 2015; Sanderson et al., 2023). In areas where
isochrones are strongly disrupted by (past or present) enhanced flow, extracting ILCI or isochrone-slope
products from the radiostratigraphy (as introduced in Sect. 4.3) has helped to complement reconstructions
of past or present ice-flow dynamics (e.g., Karlsson et al., 2012; Bingham et al., 2015; Holschuh et al., 2017;
Ashmore et al., 2020; Luo et al., 2020; Sanderson et al., 2023). In some cases, sequences of folded isochrones
have been observed beneath sequences of conformable isochrones, indicative of a past sudden change from
fast to slow ice flow (e.g., Conway et al., 2002; Siegert et al., 2013; Kingslake et al., 2016). To obtain more
complex information on past ice-dynamic changes falls into the realm of applying numerical modelling, which
is taken up in Sect. 5.6.
An important outcome of most ice flow is that the ice crystals themselves develop a preferred orientation,
typically termed anisotropic crystal-orientation fabric, which may then influence the direction-dependent
propagation speed of radio waves through ice (Gow and Williamson, 1976; Robin and Millar, 1982; Fujita et
al., 1999; Matsuoka et al., 2003; Eisen et al., 2007; Drews et al., 2012; Jordan et al., 2020; Jordan et al., 2022).
Studies have reconstructed and constrained the mechanical anisotropy of ice and histories of ice deformation
by calculating the travel-time difference for IRHs across intersecting RES profiles where the radio waves have
been polarised in different directions (e.g., Fig. 8g; Ershadi et al., 2022; Jordan et al., 2022; Gerber et al., 2023;
Zeising et al., 2023). A special case of isochrone folding due to changes in ice-crystal fabric occurs at ice
divides, where upward-pointing folds termed Raymond Arches (Fig. 8h) form due to the interplay of the
strain-rate dependence of ice viscosity, which leads to stiffer ice beneath the divide, slowing isochrone
thinning down relative to the flanks (Raymond, 1983; Vaughan et al., 1999; Martín et al., 2009; Hindmarsh et
al., 2011; Matsuoka et al., 2015). The special geometry of these isochrone arches has been used to infer local
ice-flow history including the onset of divide flow (Conway et al., 1999; Kingslake et al., 2016), divide



migration (Nereson et al., 1998; Martín et al., 2009; Schannwell et al., 2019) and ice-thickness changes (Drews
et al., 2015). With stable ice-divide positions over extended periods of time, these arches can evolve further
into double-peaked Raymond Arches, as observed (Drews et al., 2013) and simulated by incorporating
anisotropy into the ice-flow models (Pettit et al., 2007; Martín and Gudmundsson, 2012; Martín et al., 2014).
In terms of efforts to trace isochrones widely across the Antarctic ice sheets, Raymond Arches have the
greatest relevance in how they affect site selection for deep ice cores that are ideally used to assign ages to
Antarctic-wide isochrones (as introduced in Sect. 4.4). The relative thinness of isochrones at the apex of
Raymond Arches implies that better resolution age-depth records reaching further back in time would be
obtained around the flanks, rather than on the apexes, of ice divides where arches are present.
**5.5 Applications of internal architecture complementary to radiostratigraphy**
The basal ice of Antarctica and Greenland is typically characterised by an echo-free or low-backscatter zone
lacking coherent layered reflections, termed an *echo-free zone* (EFZ) in early observations (Drewry and
Meldrum, 1978; Robin and Millar, 1982; Fujita et al., 1999). With modern RES systems, this zone now appears
as a basal unit in which IRHs are often warped, folded and winnowed out, and consequently lack coherent
reflections (Drews et al., 2009), but even without traceable radiostratigraphy this architecture contains useful
information about ice properties and origins. With the progressive enhancement of RES-system range
resolution, a variety of reflection sub-units distinctly standing out from the otherwise low-backscatter zone
have been identified (e.g., Fig 8i; Bell et al., 2011; Bell et al., 2014; Wrona et al., 2018; Ross et al., 2020; Lilien
et al., 2021; Franke et al., 2024). Some of these features manifest as zones with nearly continuous high
backscatter spanning several hundred metres in thickness. Some features drape over mountainous subglacial
regions (e.g., in Antarctica's Gamburtsev Mountains and Jutulstraumen drainage basin; Bell et al., 2011;
Wrona et al., 2018; Franke et al., 2024), while others build plume-like structures within the cores of englacial
folds (e.g., in northern Greenland and Antarctica's Institute Ice Stream; Bell et al., 2014; Ross et al., 2020).
These basal units are likely of different origins and exhibit different dielectric properties compared to their
low-backscatter surroundings, offering insights into potential formation mechanisms. Current hypotheses
include strong deformation on the micro-scale by ice dynamics (Drews et al., 2009), freeze-on of subglacial
water at the ice base (Bell et al., 2011; Creyts et al., 2014; Leysinger Vieli et al., 2018), and the incorporation
of point reflectors (e.g., basal sediment; Winter et al., 2019b; Franke et al., 2024), as well as ice flowing over
regions with changes in basal friction (Wolovick et al., 2014; Wolovick and Creyts, 2016) or convergent flow
(Bons et al., 2016; Ross et al., 2020). The presence of these basal units can influence the rheological
properties and fabric structure of the ice column, as well as impact the continuity of climatic records,
highlighting their significance for ice-core drilling projects and ice-flow-modelling endeavours (Bell et al.,
2014; MacGregor et al., 2015a; Panton and Karlsson, 2015).



Buried surface crevasses imaged in RES data have been used as key evidence for timing the shutdown of
Kamb Ice Stream (Retzlaff et al., 1993; Jacobel et al., 2000; Smith et al., 2002; Catania et al., 2006) and the
reorganisation of flow through Whillans Ice Stream (Conway et al., 2002). The locations and geometry of
basal crevasses formed near the grounding line (Fig. 8j) have also been used to identify previously floating
ice, and time the formation of ice rises and ice-flow reorganisation during the Holocene in Antarctica's
Weddell Sea Sector (Kingslake et al., 2018; Wearing and Kingslake, 2019).
Finally, some particularly bright isochrones have been used to constrain the timing of past volcanic eruptions
and constrain the ranges of their tephra fallout. Most such reflectors are relatively bright through chemical
signatures alone (e.g., Welch and Jacobel, 2003), but a particularly prominent isochrone, ~30 dB stronger
than other typical isochrone-reflection strengths, and thus interpreted as containing physical tephra
fragments in addition to chemical residues, was mapped and interpreted by Corr and Vaughan (2008) to
demonstrate a volcanic eruption occurred ~2000 years ago in West Antarctica and covered much of the Pine
Island Glacier basin (Fig 8k).

**5.6 Using isochrones in ice-sheet models**

Ice-flow models of different complexities comprise the foremost tools for projecting future ice-sheet and
glacier evolution (e.g., Gagliardini et al., 2013; Cornford et al., 2015; DeConto and Pollard, 2016; Seroussi et
al., 2020). Incorporating radiostratigraphic data into ice-sheet models provides a means for validation,
improves their calibration and might be essential for making more robust projections by models seeking to
constrain ice-sheet evolution over the past few centuries to the late Quaternary (Hindmarsh et al., 2009;
Leysinger Vieli et al., 2011; Holschuh et al., 2017; Born and Robinson, 2021; Sutter et al., 2021). Palaeo-proxy
records such as exposure-age dating (Brook and Kurz, 1993; Mackintosh et al., 2014; Hillebrand et al., 2021),
grounding-line reconstructions (Bentley et al., 2014; Wearing and Kingslake, 2019) or estimates of past sea-
level highstands (Dutton et al., 2015) provide invaluable snapshots of ice-sheet variability on local, regional
and continental scales (Lecavalier et al., 2023, present a state-of-the-art database), but their interpretation
remains challenging in terms of attribution of ice volume, and changes to the grounding zone and ice
elevation. Dated radiostratigraphy, on the other hand, contains detailed information on the evolution of ice
flow on the relevant timescales (as compiled for today in Sect. 4.5) and thus provides a much-refined
calibration target bridging gaps in between snapshot proxy data. Although the theoretical link between ice
flow and isochrone geometry has been established for steady tube flow of an ice sheet (Parrenin and
Hindmarsh, 2007), the general 3D and transient case remains far more challenging. In this section, we
overview recent developments in ice-sheet modelling that incorporate or exploit isochronal data from RES
surveys.

**5.6.1 Modelling past climate and ice-dynamic changes**



Radiostratigraphy is an ideal tuning target for ice-sheet models on continental, regional (catchment) and local
scales, because it inherently records the history of the ice flow as well as its response to changing climate
conditions in its geometry. As opposed to traditionally-employed tuning targets such as surface flow, ice-
sheet geometry or ice volume, which only represent snapshots of ice-sheet evolution, radiostratigraphy
provides a 3-D structure which has been formed by the transient palaeo-evolution of the ice sheet. Modelling
isochronal geometry and age is technically relatively straightforward, with the main challenge being
pervasive uncertainties in boundary conditions (e.g. climate forcing and geothermal heat flux) and the
intrinsic uncertainties of ice-sheet models due to their parameterisations of physical processes (Sutter et al.,
2021). Isochrones in RES data, age-depth profiles in ice cores and the isotopic content of ice sheets have
been modelled either by employing Lagrangian (Sutter et al., 2021) or semi-Lagrangian (Tarasov and Peltier,
2003; Clarke et al., 2005; Goelles et al., 2014) advection or isochronal models (Born, 2017; Rieckh et al., 2024).
Models that simulate stratigraphy can thus be used to explore the effects of palaeoclimate evolution on ice-
dynamic changes, such as marine ice-sheet instabilities or the evolution of ice-sheet drainage systems.
Continental-scale ice-sheet models employing approximations of the full-Stokes equations have allowed the
computation of ice flow on time scales of centuries to millions of years, albeit at the cost of resolution, which
is usually ~5–40 km (Pollard and DeConto, 2009; Golledge et al., 2015; Sutter et al., 2019; Albrecht et al.,
2020; Seroussi et al., 2020). While these relatively coarse grid sizes (compared to applications of full-Stokes
models; e.g. Zhao et al., 2018) preclude a meaningful interpretation of small-scale processes that influence
radiostratigraphy (e.g. local freezing, melting, bedrock features etc.), large-scale models have the advantage
that they incorporate the whole thermomechanically-coupled ice-sheet system and its response to changing
climate conditions. Consequently, large-scale models are also the main tools for projections of sea-level
contributions from the Antarctic and Greenland ice sheets (e.g., Goelzer et al., 2020; Seroussi et al., 2020).
The analysis of isochrones to inform on past ice flow need not be limited to the grounded parts of an ice
sheet and has been extended to ice shelves (Višnjević et al., 2022; Moss et al., 2023), ice rises (Goel et al.,
2018; Goel et al., 2024), and the ice-rise/ice-shelf system (Henry et al., 2024). In these studies, isochrones
have served as valuable resources for reconstructing both the surface and/or basal mass balance of ice
shelves and ice rises using forward and inverse modelling along the flowline (in 2D), and for investigating
rheological properties of ice rise/ice shelf systems in 3D (Henry et al., 2024). Extending this approach to
include the past ice-shelf evolution and linking the isochronal structure to its grounded counterparts remains
challenging due to the lack of tie points to dated isochrones and a lack of observable isochronal structure
across the grounding line.
**5.6.2 Model integration of isochronal data**
A range of models has been used to calculate the age-depth relationship in ice over both large and small
portions of Antarctica and compare this with existing radiostratigraphies; an exercise that can offer valuable



insights into ice-sheet processes and how these are represented in ice-sheet models (Fig. 9). When
integrating isochronal data in models, multiple factors play a role in the choice of model set up, such as the
size of the area of interest (e.g. regional or continental) and the type of flow regime present (e.g. dome,
vertical shearing, extension). Various types of flow regime are found in Antarctica, ranging from vertical
compression at domes moving to vertical shear and finally to longitudinal extension in ice streams and ice
shelves. Consequently 1D, 2D or 3D models might be the optimal choice to simulate the age or stratigraphy
of ice, with 2.5D models, i.e. 2D models that take into account some aspects of a third dimension, providing
another option (Chung et al., 2024).
1D models typically assume negligible horizontal flow, making simplifying assumptions such as a steady-state
velocity field and the local layer approximation (Waddington et al., 2007, provide guidelines on its
applicability) and have predominantly been used at domes such as Dome C (Parrenin et al., 2017; Lilien et al.,
2021; Chung et al., 2023) and Dome F (Obase et al., 2023; Wang et al., 2023), where vertical compression
dominates. Dated isochrones have been used in multiple studies to constrain 1D age-depth models of
different complexity to determine millennial-scale accumulation rates in Antarctica (e.g., Leysinger Vieli et
al., 2004; Siegert and Payne, 2004; MacGregor et al., 2009; Karlsson et al., 2014; Koutnik et al., 2016; Cavitte
et al., 2018; Zhao et al., 2018; Ashmore et al., 2020; Bodart et al., 2023; Sanderson et al., 2024) and retrieve
horizontal flow velocity from 2D isochrone architecture (Eisen, 2008). While most such studies have been
restricted to using steady-state due to temporal limitations in available data, some models have allowed for
temporal changes in boundary conditions (Callens et al., 2016; Parrenin et al., 2017; Chung et al., 2023).
3D modelling of ice-rise stratigraphy (Henry et al., 2024) has provided a step towards constraining long-term
simulations in coastal areas. The influence of model physics on this stratigraphy was first investigated in 2D
idealised studies of Raymond arches (Pettit and Waddington, 2003; Pettit et al., 2007; Martín and
Gudmundsson, 2012), with Hindmarsh et al. (2011) extending this work in 3D idealised simulations.
Modelling studies have examined the influence of Glen's flow law exponent on Raymond-arch amplitude
(Pettit and Waddington, 2003; Martín et al., 2006; Martín and Gudmundsson, 2012). This methodology has
been extended to 2D simulations of real-world ice rises and domes in coastal Antarctica with the comparison
of modelled and observed Raymond arches at ice divides (Martín et al., 2009; Hindmarsh et al., 2011; Pettit
et al., 2011; Martín et al., 2014; Drews et al., 2015; Goel et al., 2018; Goel et al., 2024).
Isochrones have also been used to estimate ice temperature on catchment- to continent-wide scales.
Because the electrical conductivity of ice varies exponentially with temperature, resulting in higher dielectric
attenuation in warmer ice (MacGregor et al., 2007), temperature variability across the ice sheets leaves a
signature in the returned power of measured radio waves. To date, studies have concentrated on using
thermomechanical ice-sheet models to improve interpretation of RES data by using modelled temperature
fields to remove attenuation effects and strengthen interpretations of bed properties based on basal



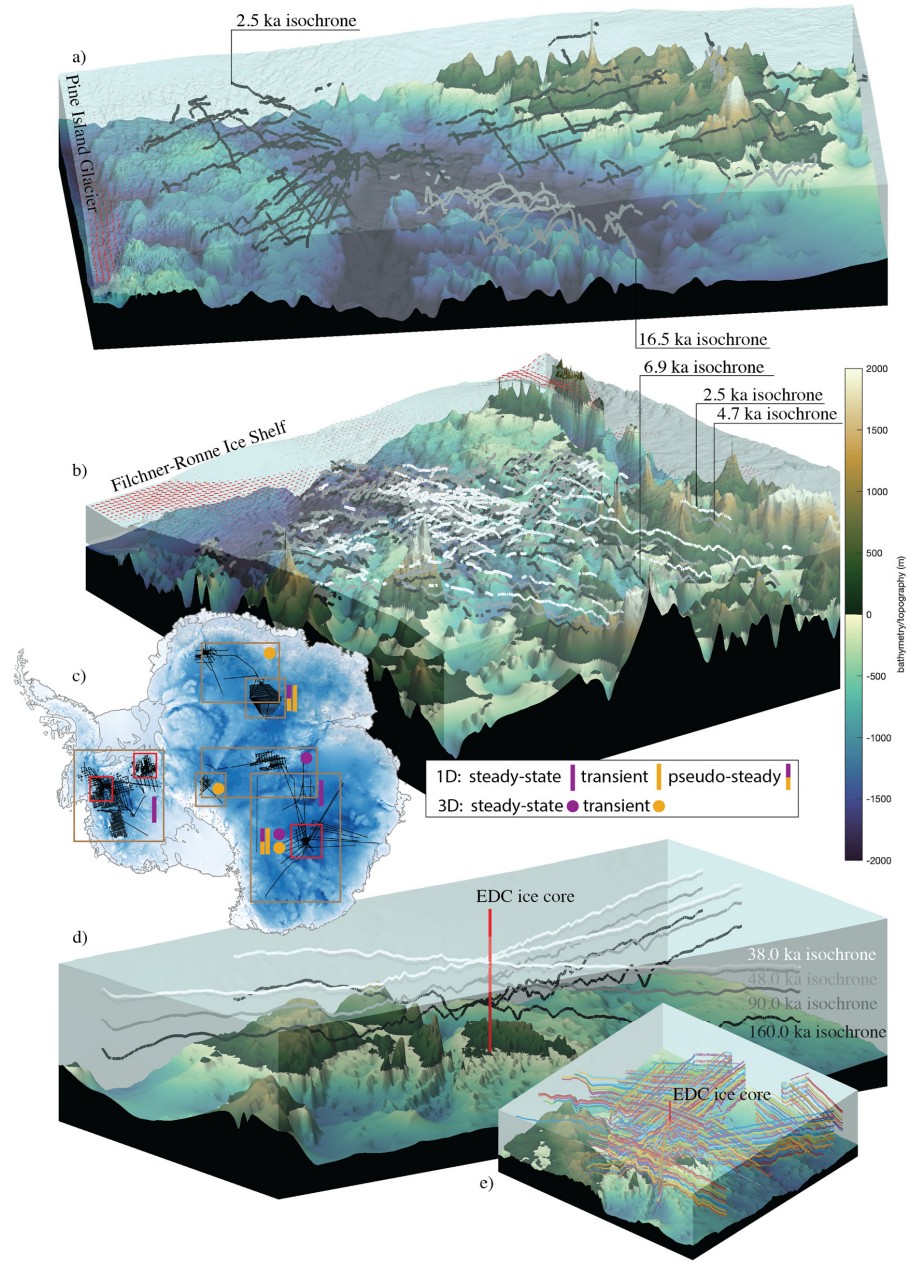

**Figure 9.** 3D visualisation of selected traced and dated isochrones in East And West Antarctica, and locations where different modelling applications have been conducted. (a) 2.5 ka (black lines) and 16.5 ka (grey lines) isochrones across the Pine Island/Thwaites Glacier catchment area (Bodart et al., 2021). (b) 2.5, 4.7 and 6.9 ka isochrones spanning Institute Ice Stream (Ashmore et al., 2020). (c) Map of Antarctic traced and dated isochrone transects (black lines) and areas where at least one modelling study is available (grey boxes); red boxes denote areas of the 3D visualisations. (d) Traced and dated (38, 48, 90, 160 ka) isochronal structure around Dome C from Winter et al. (2019a) and (e) Cavitte et al. (2021).





reflectivity (Matsuoka et al., 2012; MacGregor et al., 2015b; Chu et al., 2021; Dawson et al., 2022). This
approach assumes that thermomechanical models can estimate the ice temperature field to high confidence.
Additionally, 1D age-depth models that incorporate a thermomechanical component (Parrenin et al., 2017;
Passalacqua et al., 2017; Obase et al., 2023) have been used to infer basal melt rates in Antarctica close to
domes. Temperature modelling, however, can be challenging in fast-flowing areas where heat production by
viscous dissipation is substantial, such as along shear margins or ice streams. As efforts to reduce ambiguity
in the direct inference of temperature from RES reflection strength develop, it will become possible to
assimilate RES measurements of temperature to improve model performance, as has been done with other
direct and indirect observations of subsurface temperature (Pattyn, 2010; Van Liefferinge and Pattyn, 2013).
While a combined evaluation of model temperature and velocity data from RES data has been performed
qualitatively (Holschuh et al., 2019), there is a growing desire to incorporate *both* radiometric and structural
information in a formal modelling framework.

**6 Future directions**

In this review, we have considered how the internal architecture of the Antarctic ice sheets, and in particular
their radiostratigraphy, is increasingly being exploited to elucidate ice and climate history. The ultimate aim
of these endeavours is to constrain in ever finer detail the rates, locations and underlying processes of past
ice-sheet changes in response to climate forcing. This is crucial to inform and reduce uncertainties in models
projecting future ice-sheet changes and concomitant global sea-level rise. Yet, despite the progress reported
above, Antarctica's internal architecture remains an underutilised resource for this purpose. In this final
section, we set out recommendations for future research activities to be underpinned by an expanded and
accessible database of Antarctica's internal architecture. Firstly (Sect. 6.1), we present a pathway towards
expanding the volume of radiostratigraphy across Antarctica towards the goal of building a 3-D age-depth
model of the ice; secondly (Sect. 6.2), we set out a number of future science challenges that a comprehensive
database of Antarctica's englacial architecture can help to address; and finally (Sect. 6.3), we make some
recommendations for community actions to facilitate the delivery of these goals.

**6.1 Pathway to expanding Antarctic radiostratigraphy**

We have identified throughout this review a clear need to expand significantly the traced radiostratigraphy
across the Antarctic ice sheets, covering both more area and a greater depth range through the ice. To
achieve this requires the following steps:

***6.1.1 Numerical modelling to guide where radiostratigraphic constraints are most needed***

We recommend that future targets for tracing radiostratigraphy across different regions of Antarctica, from
existing RES data or guiding new RES surveys, are informed directly by the needs of the ice-sheet modelling



community to benchmark and constrain their models. Modelling can guide location-based suggestions (e.g.
to recover more radiostratigraphy away from ice divides and into more dynamic regions where simple model
heuristics may misrepresent englacial conditions), or require targeting of particular time periods (e.g.
targeting older isochrones that could advance understanding of glacial-interglacial transitions, amongst
others).

### 6.1.2 Systematic assessment of the potential of existing data for tracing radiostratigraphy

For this review, we have compiled the spatial coverage of existing published RES data across Antarctica that
have high-quality (GNSS) navigation and were acquired digitally, and often coherently (Figure 3l). In principle,
this demonstrates the present coverage of RES data from which radiostratigraphy could be extracted and
mapped, and indicates that RES datasets range and interconnect widely across both the East and West
Antarctic ice sheets. While this presents a positive message of the potential for pan-Antarctic tracing of
radiostratigraphy, whether and how much radiostratigraphy *can* be extracted so widely across the ice sheets
from all of these profiles remains unknown. Not all of the RES tracks necessarily contain traceable
radiostratigraphy, for reasons that range from inherent RES-system limitations upon data acquisition,
decisions made in the processing of the data that are available (see Sect. 4), to the presence of physical
phenomena in the ice that disrupt radiostratigraphy or steeply sloping basal topography that makes
isochrones too steep to be traced (Sect. 5).
A community effort is therefore required to investigate the full potential for mapping radiostratigraphy
through these existing datasets. A useful first step, which was beyond the scope of this paper, would be to
apply the ILCI to all of the modern datasets presented in Figure 3l to assess their viability for tracing
isochrones across different regions, i.e., to produce a more comprehensive version of Figure 6 expanded to
all the datasets discussed in Sect. 3.

### 6.1.3 Reprocessing of existing datasets to accentuate internal architecture

While the visibility of internal architecture is partly determined by the initial acquisition parameters and
varies across Antarctica (Sect. 3), the information visible in RES data is also influenced significantly by the
processing applied to the data *after* they have been acquired (Sect. 4.1). Where the raw data exist, the data
can be reprocessed, which may significantly enhance the value of some existing datasets for tracing their
radiostratigraphy. For much of Antarctica's RES data, the only processing that has been applied was
implemented to emphasise and pick the bed echo. In some cases, the same processing accentuated
radiostratigraphy in parallel but, in others, it has suppressed the imaging of isochrones or induced artefacts
in the radargrams that have hampered or precluded any tracing of radiostratigraphy. Therefore, where
existing data lack distinct isochrones in locations identified by numerical modelling as optimal candidates for
radiostratigraphy, we recommend, where feasible, firstly reprocessing the raw data to enhance internal



architecture. Such an initiative is currently being trialled as part of the Open Polar Radar project using AWI,
BAS and USA-acquired RES data across Antarctica (Paden et al., 2021).

### 6.1.4 New data acquisition

Importantly, new RES data for radiostratigraphic constraints need only be acquired where the processes
described above have highlighted that existing data cannot provide the radiostratigraphic constraints
required by modelling applications. Such areas will fall into three categories:
(a) Regions that are still unsurveyed or undersurveyed. Clear examples of this situation, from Figure 3l,
comprise data gaps > 100 km wide in East Antarctica in Enderby Land; between South Pole and Vostok;
and between Wilkes and Kemp lands; and we also note that the Filchner-Ronne Ice Shelf does not have
dense survey cover.
(b) Regions where RES surveys have occurred but where the existing data – even after reprocessing – do not
contain any internal architecture. These regions typically comprise those last surveyed by RES several
decades ago with less sophisticated RES systems. From Figure 3, we identify the Siple Coast region of West
Antarctica as one such data gap. Although this region was intensively studied and surveyed during the
1980s and 1990s, its last major RES surveys predate widespread use of coherent RES systems.
(c) Regions where RES surveys have occurred but where the existing data – even after reprocessing – contain
some internal architecture, but which does not meet modelling needs. Likely scenarios here are that age-
depth information is needed at finer resolution than is retrievable in the existing data, or there is a
requirement to recover radiostratigraphy deeper into the ice than has been imaged by the existing survey.
This situation is common amongst existing datasets that were acquired for projects focussed on other
scientific priorities. For example, where some airborne RES datasets have been acquired in combination
with potential-field data (gravity and magnetics), the requirement to fly the aircraft at a stable elevation
has sometimes led to poor-quality radiostratigraphy where the range from aircraft to ice surface was too
large.
These cases should fundamentally guide the locations, nature and platforms of any new RES data acquisition
for internal architecture. As reviewed in Sect. 3, modern airborne RES systems and processing algorithms are
adept at detecting multiple isochrones over large regions. In some cases, such as through regions of complex
topography, complex flow dynamics or a requirement for very fine resolution of isochrones over regional
scales, ground-based RES systems that can typically sound more IRHs and deeper into the underlying ice may
still represent the optimal tool and justify the resources required to emplace deep-field parties. However,
uncrewed aerial vehicles capable of carrying RES systems (Arnold et al., 2020; Teisberg et al., 2022), when
routinely operationalised, may offer a cheaper and safer solution over remote and challenging terrains.





***6.1.5 Advances in deep learning to expedite the extraction of internal architecture from RES data***
As reviewed in Sect. 4, all of the present radiostratigraphy mapped across Antarctica (Fig. 7) has been
generated in the absence of a fully automated isochrone-picking algorithm. Although substantial progress
has been made, the need for frequent manual intervention has slowed the generation of pan-Antarctic
radiostratigraphy. The greatest promise for a step-change in our ability to trace radiostratigraphy significantly
faster lies in the application of deep-learning methods to the challenge. As we discussed in Sect. 4.2, deep
learning has so far only been implemented to tracing shallow isochrones in the first few hundred metres of
ice, which are typically more continuous over many 100s of km. Tracing isochrones deeper in the ice column
is challenged by IRH fading, unconformities, and/or merging and splitting of isochrones as ice flows over or
around large bedrock obstacles. However, the significant volume of traced radiostratigraphic data now
assembled to date across Antarctica (Fig. 7) can now contribute training data to facilitate the advance and
wider application of deep learning to tracing Antarctica's deeper isochrones.
**6.2 Recommendations for future scientific deliverables using internal architecture**
***6.2.1 Identification of optimal areas for retrieving new palaeoclimate records***
As outlined in Sect. 5.1, Antarctica's deep ice cores have provided invaluable palaeoclimate records from
both West and East Antarctica and yet there remain two outstanding directives in the quest for augmenting
these existing datasets. One, presently the primary focus of the SCAR IPICS *Oldest Ice* programme, is to
identify where a potential climate record extending further back in time than Antarctica's current record
(back to ~800,000 k.a. from Dome C; Bouchet et al., 2023) can be sampled. This would address the substantial
unknown of whether Antarctica's ice holds a direct continuous record of the mid-Pleistocene transition
switch from 41-kyr to 100-kyr glacial-interglacial cycles that is inferred to have occurred between ~1.25-0.8
M k.a. from marine-sediment oxygen-isotope records (Hays et al., 1976; Clark et al., 2006; Legrain et al.,
2023). A second requirement is to locate sites in the Antarctic ice sheets that preserve higher-resolution
palaeoclimate records of epochs than are currently represented in the already-sampled sites. In particular,
regions with relatively high present or past accumulation rates can potentially preserve high-resolution
climate records of the last millenia. We contend that the development of a pan-continental radiostratigraphy
could form a crucial tool for identifying most future ice-core locations around Antarctica.
We further recommend that attention is placed on tracing radiostratigraphy around Antarctica's blue-ice
zones which, as discussed in Sect. 5.1, have and can represent sites for retrieving ice older than 800 k.a.
Targeted studies on their radiostratigraphy could improve understanding of how ice deforms to produce the
sampled structures, and hence better contextualise how the ice outcropping in such regions is related to ice
buried at depth in interior Antarctica.



These initiatives may be complemented by the strategic deployment of rapid-access drilling techniques that
could be deployed, alongside intersections with ice cores (discussed in Sect. 5.1), to date and validate the
radiostratigraphy. Rapid-access drilling (e.g., Goodge and Severinghaus, 2016; Rix et al., 2019; Goodge et al.,
2021; Schwander et al., 2023) can provide borehole access into the ice for deploying sensors to record
physical characteristics that correlate with RES isochrones (IceCube Collaboration, 2013; Goodge et al., 2021;
Schwander et al., 2023). Additionally, rapid-access drilling allows direct sampling of ice that can be used for
radiometric-age dating that can validate the radiostratigraphy (e.g., Bender et al., 2008; Rowell et al., 2023).
A dedicated programme of rapid-access ice drilling coordinated with *AntArchitecture* could therefore both
help to validate radiostratigraphic age-depth models, and provide a relatively quick and cost-effective
methodology for targeting potential future sites for both vertical and horizontal ice coring.
***6.2.2 Reconstruction of surface mass balance – millennial timescales***
In Sect. 5.2, we discussed that tracing deep (>200 m below the ice surface) isochrones across the Antarctic
ice sheets enables reconstruction of changes in surface mass balance over the past several millenia. While
the few existing studies have mostly focussed at or near ice divides, where horizontal flow and its associated
complexities can mostly be neglected, an expanded pan-continental radiostratigraphy that more
comprehensively spans and connects all of Antarctica's central divide regions will enable these simple
applications to be expanded, and can provide a spatially widespread record of how surface mass balance has
varied regionally at millennial timescales. Such a record would help us to understand the pervasiveness of
synoptic snow-accumulation patterns (e.g., Le Meur et al., 2018; Pauling et al., 2023), and could inform
scenarios of future plausible surface-mass-balance variability to be incorporated into model projections (see
Lenaerts et al., 2019, for a review). In turn, such refined surface-mass-balance reconstructions would greatly
improve the climate forcings employed by palaeo-ice-sheet-modelling studies and increase confidence in
their conclusions.
***6.2.3 Reconstruction of surface mass balance – historical timescales***
To reduce uncertainties in near-term (i.e., ~next 200 years) projections of Antarctica's future evolution, and
thereby improve global sea-level projections, there is a critical need to constrain further the regional climate
models (e.g., Pratap et al., 2022) that are fundamental to forcing ice-sheet models. Important validation for
these models comes from the historical record provided primarily by ice cores, but also by radiostratigraphy
sounded in the upper few 100 m of the ice sheet, hereafter termed *shallow radiostratigraphy*. Neither this
review, nor the *AntArchitecture* community to date, has focussed on shallow IRHs. However, the majority of
RES surveys depicted in Figure 3 also detected shallow radiostratigraphy, and many additional surveys have
been undertaken over the past decades across Antarctica using a range of airborne and ground-based
platforms that focussed on detecting shallow isochrones, often for local, but sometimes also for more
regional, scientific applications (e.g., Medley et al., 2013; Medley et al., 2014; Konrad et al., 2019; Kowalewski





et al., 2021; Cavitte et al., 2022). We therefore propose that an important future deliverable should be a "shallow" of pan-Antarctic radiostratigraphy complementary to the deeper version that has primarily formed the focus of this review. In parallel with the techniques and philosophy we have discussed for dating deep isochrones across Antarctica, shallow radiostratigraphy can be dated from intersections with shallow-ice-core records; and the product could be progressively refined by using it to identify where future shallow-ice cores should be drilled to provide finer dating control. It is likely that the overall task of tracing shallow isochrones across Antarctica could benefit from the application of machine learning to isochrone tracing sooner than for deeper isochrones, as the former are typically less disrupted by ice dynamics and are more continuous. Indeed, shallow isochrones have already been traced with deep learning with some success in several studies (e.g., Dong et al., 2021; Rahnemoonfar et al., 2021; Yari et al., 2021).

### *6.2.4 Estimate geothermal heat flux from radiostratigraphy*

The studies mentioned in Sect. 5.3 speak to the significant potential for Antarctica's radiostratigraphy to be used as a resource for constraining variations to the continent's geothermal heat flux, which remains enigmatic (Burton-Johnson et al., 2020). As exemplified by Fahnestock et al. (2001b) across the Greenland Ice Sheet, and more locally in Antarctica by Jordan et al. (2018), it is possible to quantify basal melt with isochrones by calculating how much melting is required to draw isochrones down towards the base. However, the relationship between isochrone geometry and basal melting is complex, multi-dimensional and partly controversial (Leysinger Vieli et al., 2007; Carter et al., 2009; Bons et al., 2021; Wolovick et al., 2021b; Wolovick et al., 2021a). For a continental-scale application of this technique, a more detailed pan-Antarctic radiostratigraphy is needed. The optimal data product to invert for geothermal heat flux would be the most widespread tracings of the deepest undisrupted isochrones across the ice sheets, which is challenging because deeper isochrones are harder to image and significant drawdown of isochrones where basal melting is high can prohibit widespread tracing (e.g., Ross and Siegert, 2020). Nevertheless, there is significant potential to use deep isochrone geometry as further calibration for numerical models seeking to invert geothermal heat flux (Pattyn, 2010; Van Liefferinge and Pattyn, 2013; Burton-Johnson et al., 2020).

### *6.2.5 Comprehensive mapping of basal-ice units and deep-isochrone geometry*

In Sect. 5.5, we noted that in some regions of the Antarctic ice sheets, RES data indicate that the deeper ice has distinctive physical characteristics compared with the ice above, i.e., where this deeper ice obscures or precludes imaging of IRHs, and where distinct basal-ice units exist around which the overlying IRHs have become folded or warped. An improved understanding of the distribution of these features across Antarctica is important for several reasons. Firstly, it would identify where deep-ice palaeoclimate records would be compromised by ice deformation or basal melting, thus critically informing ice-core site identification. Secondly, it would act as an observationally-informed broad-scale indicator of which areas of the ice sheet are prone to basal melting and hence inform mapping of geothermal heat flux. Thirdly, it would provide





information towards a better understanding of how the rheology of Antarctica's ice varies, what are the causes of this variation, and how these effects impact on Antarctica's ice dynamics. Some of these issues would be informed by some specific rapid-access drilling into basal-ice units, and a comprehensive mapping exercise of basal-unit distribution would inform which targets might be most easily accessed. In addition to mapping basal units themselves, a complementary deliverable could be to map the degree to which deep-ice radiostratigraphy follows or diverges from the ice-bed interface across Antarctica. This exercise would inform modelling aimed to deconvolve how much isochrone geometry is affected by basal topography versus ice dynamics versus basal melt. This, in turn, will better inform projections of the ice sheets' future with radiostratigraphic constraints.

### 6.2.6 Advance knowledge of volcanic activity and fallout across Antarctica

Given that most isochrones traced across the Antarctic ice sheets manifest changes to acidity, and that some of the brightest have been linked to precipitated fallout from volcanic eruptions within and beyond Antarctica, there is significant potential to use isochrones across Antarctica more comprehensively to trace the spatial distribution of volcanic fallout from the numerous past eruptions that have been identified by chemical analyses of Antarctica's ice cores (Narcisi and Petit, 2021). Despite many tephra and cryptotephra (microscopic layers of volcanic ash) having been detected in Antarctica's ice cores, few have explicitly been traced widely beyond the ice cores using radiostratigraphy, and most isochrones that have been linked to past volcanic events have been used as time markers for other purposes, e.g. calculating past accumulation, rather than having been traced to focus on the origins and properties of the volcanic events themselves (e.g., Jacobel and Welch, 2005; Bodart et al., 2023). There is therefore significant potential, already with existing data, to use Antarctica's radiostratigraphy to trace the geographical distribution of volcanic fallout from numerous eruptions that have been detected in ice-core records, and this information may be used to help trace further the origins and nature of past eruptions beyond that which can be gleaned solely from the ice-core chemistry. This objective would complement the ongoing activities and recent recommendations for future research on volcanism presented by the SCAR AntVolc group (Geyer et al., 2023).

### 6.2.7 Development of a new model benchmark for the Antarctic ice sheets

As reviewed in Sect. 5.6, the vast majority of ice-sheet models presently employed for ice-sheet reconstruction and future projections are initialised with present-day snapshots of the ice-sheet state (e.g., surface velocity, ice thickness). An Antarctic-wide radiostratigraphy would provide a much better initialisation and tuning target for ice-sheet models, as it inherently records both ice-flow history and the ice sheet's response to changing external forcings (e.g., atmospheric and ocean conditions) – all within a tangible set of physical horizons that can be reproduced by existing models. The development of an Antarctic-wide radiostratigraphy is therefore a primary scientific objective for SCAR's *AntArchitecture* community.



**6.3 Community actions**

The greatest challenge for attaining the deliverables described above is how to foster and maintain engagement between scientists working across numerous different disciplines and operating at institutions spread across Earth. Even within the scientific community who self-describe as RES, radar, or even radioglaciology specialists, this challenge is innate. As we have reviewed, the history and ongoing practices of Antarctic RES surveying encompass multiple agencies whose foci are typically on medium-term projects of a few years' duration. The intent of this review was to communicate to a wider audience (both within and beyond the radioglaciology community) the baseline availability and potential of the present archive of existing RES data spanning both East and West Antarctica's ice sheets, and to showcase their value for tackling major science questions concerning Antarctica's ice and climate history and future.

A major challenge to greater progress in the study of Antarctica's internal architecture has been the lack of a common framework for archiving RES data and metadata between different operators and potential users. The establishment of the FAIR (Findable, Accessible, Interoperable, and Reusable; Wilkinson et al., 2016) data-exchange guidelines has provided a clear framework making possible the release of RES data in open-access repositories, facilitating open-access releases of some of the datasets discussed in Sect. 3. These releases have been accompanied by interactive data portals and FAIR-compliant data standards, including rich metadata relating to the acquisition, processing and quality of the data, and provide examples for releasing further data in the future. We recommend that the next significant community data focus should be on developing common protocols for processing RES data, formatting and sharing raw data files, and in some cases reprocessing existing data to facilitate much greater interoperability of the data moving into the future. This recommendation falls into the remit of the Open Polar Radar project currently being trialled with AWI, BAS and USA-acquired RES data (Paden et al., 2021) but, specifically with regards to publishing and sharing future radiostratigraphy datasets, there remains a need to set a common standard. We suggest a standardised structure here in Appendix 1.

A core principle moving forwards with our science must also be on improving sustainability, given the significant resource and carbon impact of using aircraft and establishing deep-field camps in Antarctica. When proposing new Antarctic RES acquisition, we suggest that it first be demonstrated that it is needed, following the procedures laid out in Sect. 6.1. Although crewed airborne and ground-based RES platforms currently presently continue to provide the most reliable options, where new data are clearly needed n pathways for improving the sustainability of data collection are opening up with the development of uncrewed aerial vehicles capable of hosting RES systems (Arnold et al., 2020; Teisberg et al., 2022).

Finally, we call for continued efforts to build and enhance the inclusion and diversity of researchers involved in acquiring and analysing RES datasets towards understanding better Antarctica's past and future. This paper has benefitted immeasurably from including perspectives from authors spread across the world, navigating



different stages of their careers, and identifying as different genders, ethnicities, nationalities and religions;
and from including the expertise of field- and data-focussed scientists in the same space as the expertise of
practitioners whose focus is on applying the data and integrating them into numerical models. We conclude
by reiterating our core scientific ambitions for *AntArchitecture* above: to build a pan-Antarctic database of
isochrones that are accessible, sustainable over the long term, and useful for multiple scientific applications
across multiple users, for example ice-sheet modellers and the substantial ice-core community. Alongside
this, and of equal importance, the community that is active both in acquiring and analysing Antarctica's
internal architecture must continue to diversify.
**Author contributions**
The paper was jointly written by RGB, JAB, MGPC, AC, RJS and JCRS (the lead-writing team). All co-authors
contributed ideas, perspectives and edits. The review was conceptualised by RGB, OE, NBK, JAM, NR and DAY
as a deliverable for the SCAR *AntArchitecture* 2018-2022 Action Group. RGB coordinated the writing process.
DWA, RGB, JAB, AB, MGPC, WC, OE, NH, NBK, MRK, GJMCLV, JAM, EJM, EM, CM, FP, NR, JCRS, KW & DAY
made significant contributions to first draft compiled during the covid-19 pandemic in 2020-21, forming the
framework for the current version handled by the lead-writing team since 2023. Original figures were drawn
by KW, NBK & JAB (Fig. 1), DWA (Fig. 2), RGB (Fig. 3), SF (Fig. 4), JAB (Fig. 6 & 7), MGCP & RJS (Fig. 8), and JCRS
(Fig. 9), and Table 1 was assembled by JAB. Prior to submission, DWA, AB, RD, JWG, MRK, CM, FN, SVP, DMS,
TOT, XC & XT provided substantive edits; SF, VG, ACJC, AH, BHH, FMO, TR & SY led detailed reviews of each
section of the manuscript which shaped further edits; and OE, NBK, GJMCLV, JAM, FSLN, NR, RS, MJS & DAY
contributed final checks and perspectives informing the final version of the paper.
**Competing interests**
Nanna Karlsson is Co-Editor-in-Chief, Olaf Eisen is Advisory Editor, and Reinhard Drews, Joseph MacGregor,
Elisa Mantelli, Carlos Martín and Johannes Sutter are Editors of *The Cryosphere*.
**Disclaimer**
The views and opinions expressed here are those of the author(s) only and do not necessarily reflect those
of the European Union or the European Research Council Executive Agency. Neither the European Union nor
the granting authority can be held responsible for them.
**Acknowledgements**
This research is a contribution to the Scientific Committee for Antarctic Research's *AntArchitecture* Action
Group, and we thank members of SCAR's Physical Sciences and Geosciences Divisions for ongoing support of



the group since 2018. All of the UK-based authors acknowledge research funding support from the UK Natural
Environment Research Council, including Doctoral Training Scholarships to CJN (Edinburgh E4), RJS (One
Planet) and HD (SENSE). JCRS, JAB, AH, and VV acknowledge funding from the Swiss National Science
Foundation (grant no. 211542). MGPC is a postdoctoral researcher of the FRS-FNRS. AC, OE, EM and FP
acknowledge funding from the European Union: AC, OE and FP via the Horizon 2020 Marie Skłodowska-Curie
grant agreement no. 955750 (DEEPICE), EM from European Research Council Starting Grant 101076793. JAM,
DAY, TOT, SY and SS acknowledge support from the US National Aeronautical and Space Administration; TOT
was supported by a NASA FINESST Grant (80NSSC23K0271) and the TomKat Center for Sustainable Energy.
DAY, BHH, NH, MK, EJM, DMS, SY, MR and SS acknowledge funding from the US National Science Foundation:
for DAY, NH, MK, SY and SS through the Center for Oldest Ice Exploration, an NSF Science and Technology
Center (NSF 2019719); for DAY, SY and SS additionally from Earthcube (NSF 2127606); for BHH from an Office
of Polar Programs Postdoctoral Research Fellowship (NSF 2317927); for EJM from Geosciences Open System
Ecosystem Award NSF 2324092; for DMS from Office of Polar Programs Award NSF 1745137; and for MR
from BIGDATA (IIS-1838230, 2308649) and NSF Leadership Class Computing (OAC-2139536) awards. DAY, SY
and SS were also supported by the G. Unger Vetlesen Foundation. This paper is University of Texas Institute
for Geophysics contribution ####. AB and TR acknowledge funding from the Norwegian Research Council
Grant 314614 (Simulating Ice Cores and Englacial Tracers in the Greenland Ice Sheet). RD was supported by
an Emmy Noether Grant from the Deutsche Forschungsgemeinschaft (DR 822/3-1). SF was funded by the
Walter Benjamin Programme of the German Research Foundation (DFG; project number 506043073). ACJH
is supported by the Wallenberg Foundation (KAW 2021.0275). FMO acknowledges support from the German
Academic Scholarship Foundation. XC (Grant 42376253) and XT (Grant 42276257) were supported by
National Natural Science Foundation of China. CFD was funded by the Natural Sciences and Engineering
Research Council of Canada (NSERC; RGPIN-03761-2017) and the Canada Research Chairs Program (950-

1290   231237).



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



**Appendix: Suggested standardised structure for the publication of traced IRHs across Antarctica**

For publishing future radiostratigraphy datasets, we recommend scientists to follow the structure and naming convention specified in Table A1 for the first ten columns, after which additional columns may be added at the discretion of the scientists.

In the metadata, we recommend that authors also provide at least the following information:

(a) Name(s), version(s) and frequency of RES system(s) used.

(b) Value for speed of radar wave in ice used to convert IRH depths to metres below the ice surface.

(c) Value for any firn correction applied.

(d) The coordinate system(s) used following the World Geodetic System 1984 datum and appropriate projection (i.e., EPSG:3031 for Antarctica).

(e) If applicable, the type of radar product (e.g. waveform) on which the IRHs were traced.

(f) The uncertainties associated with either the IRH age or depth based on RES system resolution and IRH picking, amongst others. Ideally, if the metadata vary throughout the dataset, then such information should be attached to each data point as additional columns to those shown in Table A1.

(g) The source of age control (i.e., ice-core age scale, model).

Additional information may also be added to the metadata, such as the type of processing used to extract the IRHs (if different from the processing used to trace the bed); the distance in the along-track direction along the RES transect for each data point; a flag number indicating whether the ice thickness, surface and bed elevations come directly from the along-track radar or from an interpolated gridded product, if applicable; the spatial resolution (or spacing distance between each data point); the dating method (s) used to provide an age for each IRH; and the type of software and tools used to pick the IRHs. Missing values in the float data should be set to NaN and specified in the metadata. We also recommend the use of open-access and FAIR data formats for storing the data, such as CSV or tabular data file (or netcdf if CSV or tabular data file is not suitable) where metadata can be easily embedded together with the data. Finally, we recommend scientists to publish their data in open-access repositories alongside the paper publication, with a DOI that can be linked back to the original paper. Together, these suggested protocols will ensure the longevity of the data products for future applications and enable faster retrieval thereof, particularly with regards to the large data volumes expected from automatic IRH tracking algorithms in the future.





**Table A1.** Suggested standardised structure for the publication of IRH datasets associated with the
AntArchitecture community effort following FAIR data standards.
(For EGUsphere formatting, this 12-column table is presented across two rows.)
Table rows 1-6:

| Line ID or transect name | Trace timestamp (GPS time) | Longitude (decimal degrees) | Latitude (decimal degrees) | X coordinate (EPSG:3031; metres) | Y coordinate (EPSG:3031; metres) |
|---|---|---|---|---|---|
| | | | | | |


Table rows 7-12

| IRH name | IRH (two-way travel-time through ice only) | IRH depth below ice surface (metres) | Ice thickness (metres) | Surface elevation (metres a.s.l.) | Bed elevation (metres a.s.l.) |
|---|---|---|---|---|---|
| | | | | | |
