# Peer review of "Review Article Antarctica's internal architecture: Towards a radiostratigraphically-informed age–depth model of the"

_EGUsphere, 2024_

## Author Comment (AC1)

**Response to Referee #1 comments on:**

**EGUSPHERE-2024-2593 Review Article: Antarctica's internal architecture: Towards a radiostratigraphically-informed age–depth model of the Antarctic ice sheets**

**14 February 2025**

The paper has a clear structure, as it includes an introduction, a detailed review of relevant datasets, a discussion of methodologies, and a forward-looking conclusions. Each section logically follows from the previous one. Motivations are well stated, even though I would expect a dedicated sub-section named "motivation" inside the introduction. The abstract is concise, but it could be more engaging to summarize key findings and implications, as well as invite the readers to continue the reading. The clarity of the manuscript is adequate, however, in larger sections the readability is a bit lacking. The main point is in section 3: I suggest shortening the description of each subsection and consider summarizing with a table encompassing e.g. data provider, system name/type, key regions surveyed and/or coverage and relevant notes regarding each dataset or a table to collect the dataset grouped by key regions areas. This would allow readers to quickly grasp key distinctions between datasets.

*The advice given here regarding the presentation of the abstract and Section 3 is consistent with suggestions also made by Referee #2, and we take the point that the presentation in both cases can be made more engaging and efficient. We will revise both these sections accordingly.*

Regarding section 4, I assume it is related to figure 4. I suggest adding the reference to figure subsection at the beginning of each text subsection, e.g. 4.1 Pulse compression, filtering, and image focussing for optimising IRH tracing – fig.4b. So, in sections like the introduction and dataset descriptions, sentence structures are sometimes complex. I advise shortening some sentences to enhance readability.

*Line 483 had indeed clarified upfront in Section 4 that the section is supported by Figure 4. However, we had not added further references to sections of Figure 4 at the outset of each text subsection, and agree this is a helpful addition.*

Considering the scientific quality, the purpose of the work is clearly articulated, reflected in an adequate methodology, and its achievement compellingly underpinned by the evidence presented with the methods and techniques valid and suitable. In addition, the paper encompasses a robust range of references, except for some minor exceptions described below, the bibliography is sufficient and good.

*We appreciate this overall assessment. Thank you.*

Technical comments:

Figure2a: maybe consider adding an arrow to highlight the bedrock reflection

*We will add this.*

Figure4c: typo "differentiation"

*We will correct this.*

Figure7: it is hard to distinguish light green and yellow colors in panel b and c

*We will amend this.*

Figure8: I really like this figure, but may I suggest removing arrows and place the letters close to the area where they are referring to? e.g. put a small (f) close to "layer folding"

*Within the team of authors opinion is split between this suggestion and retaining the arrows; we will consider the suggestion.*

Figure9: Even though I appreciate 3D views, this one is difficult to read. I suggest rethinking this figure in 2D, using colours for the third dimension.

*We agree that 3D depictions as attempted here can be confusing. We will try to improve this visualisation to make it more accessible or alternatively will switch to a 2D representation.*

L195: typo "form"

*We will correct this.*

L.232-243: just to stay coherent with the writing style, you could add some references here, as it seems the only paragraph without references.

*Section 3 will be reworked and shortened in line with advice above and also from Referee #2. Should a form of this paragraph remain, we take the point, although the intention with all paragraphs from Lines 207 to 243 was simply to set the context that we will proceed through presenting airborne and ground-based RES surveys in the section. We only referred to Drewry/Frémand et al. in Section 3 Paragraph 1 to direct readers to broad overviews of Antarctic RES history and Medley et al. (2014) in Section 3 Paragraph 2 to direct readers to a single example of shallow RES data (that we do not revisit in the paper). We consider that the paper gives adequate reference to multiple papers on airborne and ground-based RES data throughout the rest of Section 3.*

L.244-264: I suggest shortening this subsection, sticking only to the difference between analogue-digital and coherent-incoherent.

*This section will be revised in line with both referees' suggestions.*

L.413: delete as it is a repetition of the line above

*This will be deleted if the subsection remains as part of the overall revision of Section 3.*

L.457-470: how about to replace the lines with a dotted list of the other institutions? Or another table just for them?

*This section will be revised in line with both referees' suggestions.*

L.488: I think this explanation could be placed in paragraph L244-264.

*Referee #2 has also made a similar point, and in response we propose to rework the previous Section 4.1 into a revised and shortened Section 3.*

L.580: I understand that focusing on processing details is not the focus, but when you talk about migration, it is straightforward to ask about velocity estimation, I think that a few lines should be added.

*This refers to Line 508. We take the point and will add some extra lines here.*

L.588-563: you speak about two approaches, but approach (a) has just one reference, could you provide more references? Otherwise, I would not define it as a "main approach", just a different one from the commonly used.

*This refers to Lines 558-563. We will remove "main" from Line 558.*

L1043: can you say that this is also a priority list?

*It is not necessarily a priority list so we prefer to leave this unchanged.*

Final comment: I recommend this manuscript to be published after some minor technical corrections, which are related to editing and enhancing readability, since the scientific quality of this work is clearly evident.

*We are grateful to the referee for their time spent reading the manuscript and formulating this considered advice.*

---

## Author Comment (AC2)

**Response to Referee #2 (Alan Aitken) comments on:**

**EGUSPHERE-2024-2593 Review Article: Antarctica's internal architecture: Towards a radiostratigraphically-informed age–depth model of the Antarctic ice sheets**

(paper title may alter slightly as per the discussion below)

**14 February 2025**

**Referee #2 (Alan Aitken)**

I review this review article from the viewpoint of a likely user of the product rather than a technical expert in RES or radiostratigraphy. Therefore my comments focus on the accessibility of the article to support a broad user-base for the AntArchitecture data products, rather than precision in the more technical aspects of the work, that may be better assessed by others.

First of all I enjoyed reading this paper and learned a fair amount. The paper tackles an important topic and has an important message, this being that the development of a knowledge of the internal structure of Antarctica's ice is important for a better-constrained knowledge of ice dynamics and ice histories, with consequent impacts from that. I have no criticisms with respect to the content, but I think that the message might benefit from some changes to the presentation of the work to have more meaning to a broader audience

*We are grateful to Alan Aitken for this overall assessment and the approach taken to reviewing the manuscript. We are aiming for broad accessibility and so are pleased to act upon many of the suggestions.*

Main comments

1) The value of the work outside the immediate community was not always well expressed, and I think it might be good in general to seek to define this further and keep it in focus.

Through the text there tends to be a focus on work done and work to do. For me at least there was a less clear thread on the summative value of all this work. This begins with the title, which is focused on the technical outcome, but not so much its value

Perhaps a change of title might help focus the work. I leave this to the authors but my suggestion would be something like:

Building a radiostratigraphically‑informed age–depth model of Antarctica's ice architecture to resolve cryosphere change.

*We are grateful for the suggestion and will now work with the revised title:*

*AntArchitecture: Building an age-depth model from Antarctica's radiostratigraphy to explore ice-sheet evolution*

Also, the first point of confusion comes in the title...Antarctica's ice sheets are mentioned in plural in the title and early in the abstract but there is no definition of what these ice sheets are. The reader must presume this refers to mean East and West Antarctic Ice sheets but this is not made explicit until much later on in the text.

This could be made explicit, or perhaps in the early text either the singular Antarctic Ice Sheet might be better, or perhaps just 'Antarctic ice' with the sheets to be defined later as the distinction becomes important.

*We will clarify that we have used the plural to recognise that there are some glaciological distinctions between the West and East Antarctic ice sheets (and we could also have included the Antarctic Peninsula Ice Sheet).*

2) Elements of the article read like a list of achievements completed and key findings, but the synthesis of these into a broader understanding sometimes is lacking.

In particular, for section 3 I saw the value its structure with respect to historical perspective, covering differing (yet co-evolving) equipment and data processing etc, it seems a little writer-focused, rather than reader focused. Instead of going through by data providers, I think this section might be rewritten focusing on the main 'eras' of airborne RES surveying, each with different value to resolving architecture.

With some transitional work, the main 'eras' seem to be 1) non-GNSS, analogue, incoherent; 2) GNSS, digital, incoherent; 3) GNSS, digital, coherent....so this would be my suggestion for section 3.1, 3.2, 3.3, with 3.4 for ground surveys

*Referee #1 also suggested a revision of Section 3 and we will work through this accordingly. We think your suggestion sounds good, and we will aim to implement it, although reserving the right to maintain some elements of the original structure, simply through the logistics of coordinating a large and international overall writing team.*

In section 5, to connect the applications to the data better, I would recommend at the beginning of each subsection to include a short introduction to the physical premise of the application...that is, how is the phenomenon expected to be observed in RES data....for example in section 5.4 ice flow dynamics, you might say "Moving ice causes originally flat layers to deform through folding, tilting and disruption. Therefore, deformed isochrons may be analysed to interpret past ice motions"

*We will reflect on Section 5 and modify the writing as fit.*

3) Section ordering was not ideal for me to follow the concepts

For me, section 4 I think is logically precedent to section 3 if we consider the need of the reader to understand the means of measuring and defining ice architecture, before we can meaningfully consider the value of surveys done. At least, before section 3, where we learn a lot of details about different radar systems of different providers, we need some brief introduction to the main factors that impact on data quality...this is all made quite clear in section 4.1 to 4.4 though so they could be moved en-bloc. Section 4.5 could be part of section 5

*Referee #1 has also made a similar point, and in response we propose to rework the previous Section 4.1 into a revised and shortened Section 3. We propose to maintain Sections 4.2-4.5 as Section 4 (the same order) but will also work on streamlining Sections 3 and 4 for the revision.*

4) The discussion needs a more rigorous defense

Section 6 outlines the importance of the work to develop deeper and broader knowledge of Antarctica's ice. I felt for this section that while it is a good identification of future directions within the scope of the review, it lacked a framing in the context of targeted decisions for the optimal outcome.

For section 6.1 My key question is what is the (relative) cost vs value vs risk proposition for each of these given technological, logistical and financial constraints? What is the low-hanging fruit and what are the grand challenges? I would appreciate here a table with some information on what

resources each is expected to require (e.g. IP, funding, technology, logistics, infrastructure) to achieve a significant improvement, and perhaps some indication of if these requirements can be met in the next decade or so.

*We feel that the exercise of engaging in the language suggested here is beyond the scope of this review paper, but is certainly a logical extension in the forum of SCAR, science programmes and future grant applications. We think the exercise would be appropriate for a "white paper" building on this science review article.*

For section 6.2 Deliverable is not the right word for this section. For me, a deliverable is a finite and defined outcome with a specific timeframe. Perhaps outcomes?

*We will change the wording as suggested.*

In this section it is not very clear to me the mechanisms by which these might be achieved . I think to identify the scale of the collaboration needed (i.e. is it needing a few research groups or a major multinational program), and also look at these through existing collaborative frameworks (are they sufficient?), and also to look at opportunities/needs for broad new collaboration.

*Similarly to two suggestions above, we think this is more the job of a "white paper" (probably coordinated through SCAR) that can be underpinned by the science objectives set out in this paper.*

Given the length of the manuscript, to close out , I think we could do with a distinct conclusion, even just 2 paragraphs, to reaffirm the main points of the text.

*We will add this.*

Minor comments

line 72: what is the basis of these proxy records?

*These are all namechecked in Paragraph 1 so were not expanded upon here in the abstract. If we have space to expand the abstract we will consider adding some examples, e.g. cosmogenically-sampled former ice limits, offshore sediment sequences, etc.*

line 104: also sediments are often very limited on the timescales of observation, and further are quite indirect with respect to ice conditions

*We will add this nuance.*

line 106: which ice sheets East and West?  Here is a good place to introduce the challenge of a composite ice sheet with several distinct parts.

*We will introduce this concept somewhere in the early part of the paper.*

line 107-108 - can you be more specific here?

*We are not sure what is being requested here so will take no action unless required further into review/revision.*

line 115 - a point to consider here is the extent to which RES-derived architecture can be considered total architecture...perhaps a brief comment that RES can't resolve everything

*Although we take the point, we prefer to discuss it later in the manuscript. Here we want to keep the focus on introducing the term as specifically what RES sees.*

Figure 1: Axes need to be 3D to match the figure.

*We will amend this.*

line 124 - here again is the plural necessary?..if so we need some definition.

*Addressed above.*

line 160: complex as in complicated or complex as in real and imaginary numbers? perhaps avoid complex

*We meant as in real and imaginary numbers but you are correct that the technicality is not needed here and we will remove it.*

line 194: this active field of research presumably has at least one paper to cite

*We will add some.*

line 207: instead of for over, perhaps spanning

*We will adopt this suggestion.*

line 218: 'now commonly employ state of the art' is redundant. In the absence of specifics, I think improved would do

*We will adopt this suggestion.*

line 225: 'shallow' RES here refers to 100s m where before and in Fig 2 the 'shallow' sounding reached 2 km. Perhaps a different temrinology could be used to differentiate - perhaps shallow vs intermediate vs deep-penetrating RES

*We will amend this in the revised paper.*

line 249-250 - I think the brackets can go as they include a full sentence

*We will consider this.*

line 251 - 'from before' replace with preceding

*We will adopt this suggestion.*

line 252 - 'were' needs replaced with a verb - perhaps 'being' or 'becoming'

*We will adopt this suggestion.*

line 254 - 'acquiring data digitally' > 'digital data acquisition'

*We will adopt this suggestion.*

line 280 'positional uncertainty' can be singular I think?

*We will adopt this suggestion.*

line 321 - delete shallow as the depth is defined numerically

*We will adopt this suggestion.*

line 323 - 'across both west and Eat Antarctica'. I find this confusing. Are some radar setups east and west specific but PASIN is not? Is there a important difference in acquisition for East vs West Antarctica ... if it is just that surveys have been done in multiple places then I would omit this point

*We recognise the unintended obfuscation in the original phrasing and we will rephrase within the wider revision of Section 3.*

line 532 - 'density value ... of the order of several meters'. Unless I misunderstand the use of density this does not make sense

*We agree this is unclearly phrased and will amend it.*

line 549 - Here it might be useful to distinguish semi-automated and fully-automated...if that is the intent

*We will clarify automated means fully automated.*

line 560 - My question here is whether IRH brightness necessarily translates to significance for ice conditions. This is addressed later, but perhaps a brief comment here on this link between data and reality is warranted.

*We will consider this.*

Figure 8: I found this figure, while quite nice, but very complex and hard to glean any information from.  Apart from the impression that work has been done i didn't learn so much. Perhaps fewer sub-figures and more annotations would be better?

*We take the point but the figure has already been through multiple iterations through the authorship team so we prefer not to implement further substantive changes.*

line 813 - drawing parallels with structural geology, a technical question is whether the details of the fold geometry can indicate the deformation mechanism, or if it is ambiguous.

*Unfortunately with present understanding it is challenging to deconvolve which mechanisms discussed in the preceding paragraph predominate to cause the folding observed at any given location. We will add an additional sentence or phrase to clarify this.*

*Our understanding of this could improve significantly with a greatly enhanced database of Antarctica's internal architecture against which to assess models!*

line 851 - winnowed is not I think the right word...perhaps pinched out or truncated?

*We will change this.*

line 943 - I don't get the sense here of whether you mean any model might do OK or more that for each circumstance there is an optimal model...this needs rewritten to be clear

*We will revise this to be clearer.*

line 1158 - this strikes me that the outcome of this approach would more be an upper limit on heat flux but this does not negate the value

*Agreed but this does not require revised wording.*

line 1226 - This is a key step. The need also for a standardised and automated processing (as far as it possible) is also important so might be added here if relevant. A key goal I think should be to move away from 'decadal' compilations and more towards and ongoing resource...this needs automation

*We feel that the need for standardised, ideally automated, processing is already encompassed in Lines 1220-1226 so that no further additions to the manuscript are needed here. We are also on board with the vision that the database should grow in an ongoing and accessible manner rather than in decadal compilations. Our challenge is that even before then the standards for lodging data need to be agreed upon.*

*We are grateful to Dr. Alan Aitken for his time spent reading the manuscript and formulating this considered advice.*

---

## Author Response (AR1)

**EGUSPHERE-2024-2593 Review Article: Antarctica's internal architecture: Towards a radiostratigraphically-informed age—depth model of the Antarctic ice sheets**

We thank both reviewers for their valuable feedback and insightful comments, and editor Dr. Huw Horgan for valuable further guidance. This letter explains how we have revised the paper in response.

The largest change we have implemented in the revision is to reorder and condense Sections 3 and 4 which introduce how RES data are processed, where current coverage exists across Antarctica, and where datasets have so far had internal architecture investigated (typically through tracing several IRHs). In rewriting this section we also redrew the former Figure 3 as a new Figure 6, in which we hope that the new (large) panel (d) provides a clearer instant depiction of where RES data are presently available for near-future investigations. We hope that this reordering and condensing have now made the sections collectively a more accessible and valuable review for the intended audience.

Other revisions have largely been of a minor nature, and are detailed in the below responses to each referee and editor below.

To aid with editing and re-review, we provide two versions of the new manuscript, one with changes tracked from the previous version\*, and a clean version. In our responses below, the line and section numbers in our responses refer to the document with changes tracked.

- \* For the document with changes tracked from the previous version, all changes are tracked except the following, where tracked-changes made the document inaccessibly messy:
  - New section 4, introduction then subsections 4.1 to 4.3.1 inclusive have replaced the former Section 3.
  - The following figures have been replaced
    - Fig. 1 old 2-D scalebar replaced with 3-D version
    - Fig. 2 black arrow added to panel (b) to annotate location of ice bed
    - o Fig. 3 (previously Fig. 4) updated with typo "differentiation" corrected
    - o Fig. 6 represents a reworking of the old Fig. 3
    - Fig. 7 improved colourscale and also added some new profiles to the map from Franke et al. (2025; ESSD) since the previous review
    - o Table 1 updated since previous review

**Response to Referee #1**

The paper has a clear structure, as it includes an introduction, a detailed review of relevant datasets, a discussion of methodologies, and a forward-looking conclusions. Each section logically follows from the previous one. Motivations are well stated, even though I would expect a dedicated sub-section named "motivation" inside the introduction. The abstract is concise, but it could be more engaging to summarize key findings and implications, as well as invite the readers to continue the reading. The clarity of the manuscript is adequate, however, in larger sections the readability is a bit lacking. The main point is in section 3: I suggest shortening the description of each subsection and consider summarizing with a table encompassing e.g. data provider, system name/type, key regions surveyed and/or coverage and relevant notes regarding each dataset or a table to collect the dataset grouped by key regions areas. This would allow readers to quickly grasp key distinctions between datasets.

We have extensively revised and condensed the former Section 3 into the new Sections 4.1 to 4.3.1, changing the main figure (new Fig. 6) to summarise the main data details.

Regarding section 4, I assume it is related to figure 4. I suggest adding the reference to figure subsection at the beginning of each text subsection, e.g. 4.1 Pulse compression, filtering, and image focussing for optimising IRH tracing — fig.4b. So, in sections like the introduction and dataset descriptions, sentence structures are sometimes complex. I advise shortening some sentences to enhance readability.

These comments are now incorporated into the new Section 3.1.

Considering the scientific quality, the purpose of the work is clearly articulated, reflected in an adequate methodology, and its achievement compellingly underpinned by the evidence presented with the methods and techniques valid and suitable. In addition, the paper encompasses a robust range of references, except for some minor exceptions described below, the bibliography is sufficient and good.

We appreciate this overall assessment. Thank you.

Technical comments:

Figure 2a: maybe consider adding an arrow to highlight the bedrock reflection

We have added this (updated Fig. 2).

Figure4c: typo "differentiation"

This is now Fig. 3c in which this is corrected.

Figure 7: it is hard to distinguish light green and yellow colors in panel b and c

We have amended the colour scheme (updated Fig. 7).

Figure 8: I really like this figure, but may I suggest removing arrows and place the letters close to the area where they are referring to? e.g. put a small (f) close to "layer folding"

The team of co-authors gave this serious consideration but opinion was split, and in the end we have elected not to implement the suggested alterations (Fig. 8).

Figure 9: Even though I appreciate 3D views, this one is difficult to read. I suggest rethinking this figure in 2D, using colours for the third dimension.

We agree that 3D depictions as attempted here can be confusing. However, we think it is important to show more visually (than in the 2D Fig. 7 for example) how isochrone shape and continuity vary in different regions. Fig. 9 shows the contrast between isochrones being stable at domes (Fig. 9d and e) and fragmented where there is higher ice flow and/or rougher topography (Fig. 9a and b). Highlighting these differences helps to explain why there are many types of models described in the text. Therefore, we have kept the 3D representation but have simplified the bedrock topography colour scheme to improve readability.

L195: typo "form"

Corrected (Line 205).

L.232-243: just to stay coherent with the writing style, you could add some references here, as it seems the only paragraph without references.

The section has been comprehensively revised rendering this action obsolete.

L.244-264: I suggest shortening this subsection, sticking only to the difference between analogue-digital and coherent-incoherent.

These suggestions are incorporated into the overall reordering and condensing of the **new Sections 3** and 4.

L.413: delete as it is a repetition of the line above

Actioned via the Section 3/4 overhaul.

L.457-470: how about to replace the lines with a dotted list of the other institutions? Or another table just for them?

Our revision of the former Section 3 has removed this requirement.

L.488: I think this explanation could be placed in paragraph L244-264.

We agreed and this has been actioned in our revision and reordering of Sections 3 and 4.

L.580: I understand that focusing on processing details is not the focus, but when you talk about migration, it is straightforward to ask about velocity estimation, I think that a few lines should be added.

This refers to Line 508 of the former version. We are not clear exactly what the referee is expecting here. If this is related to the assumption that the aircraft travels at a constant velocity and that this velocity is known and used dynamically throughout the migration progress (which are both important components of the reference phase-shift functions used to migrate radar data; e.g., Heliere et al., 2007, Peters et al., 2007), then we believe that this is perhaps too technical a point to add here in an already very technical paragraph (as acknowledged by the referee), but we leave it up to the Editor to decide if this deserves a few sentences. If, on the other hand, this is related to the assumption that we know the true speed of electromagnetic wave through the ice as the wave travels through different medians including firn and solid ice, we briefly mention this point when discussing the age uncertainty of reflector depths (now Lines 357-358).

L.588-563: you speak about two approaches, but approach (a) has just one reference, could you provide more references? Otherwise, I would not define it as a "main approach", just a different one from the commonly used.

This refers to Lines 558-563 of the former version, and is now at Lines 300-301. Firstly, we have now added an extra reference – MacGregor et al. (2025, ESSD). Secondly, we have removed the word "main" from the sentence.

L1043: can you say that this is also a priority list?

It is not necessarily a priority list so we prefer to leave this unchanged.

Final comment: I recommend this manuscript to be published after some minor technical corrections, which are related to editing and enhancing readability, since the scientific quality of this work is clearly evident.

We are grateful to the referee for their time spent reading the manuscript and formulating this considered advice.

**Response to Referee #2 (Alan Aitken)**

I review this review article from the viewpoint of a likely user of the product rather than a technical expert in RES or radiostratigraphy. Therefore my comments focus on the accessibility of the article to support a broad user-base for the AntArchitecture data products, rather than precision in the more technical aspects of the work, that may be better assessed by others.

First of all I enjoyed reading this paper and learned a fair amount. The paper tackles an important topic and has an important message, this being that the development of a knowledge of the internal structure of Antarctica's ice is important for a better-constrained knowledge of ice dynamics and ice histories, with consequent impacts from that. I have no criticisms with respect to the content, but I think that the message might benefit from some changes to the presentation of the work to have more meaning to a broader audience

Thank you for your overall assessment and for your approach taken to reviewing the manuscript. We are aiming for broad accessibility and so are pleased to act upon many of the suggestions.

**Main comments**

1) The value of the work outside the immediate community was not always well expressed, and I think it might be good in general to seek to define this further and keep it in focus.

Through the text there tends to be a focus on work done and work to do. For me at least there was a less clear thread on the summative value of all this work. This begins with the title, which is focused on the technical outcome, but not so much its value

Perhaps a change of title might help focus the work. I leave this to the authors but my suggestion would be something like:

Building a radiostratigraphically-informed age—depth model of Antarctica's ice architecture to resolve cryosphere change.

We have changed the title to:

AntArchitecture: Building an age-depth model from Antarctica's radiostratigraphy to explore ice-sheet evolution (Lines 1-5!)

Also, the first point of confusion comes in the title...Antarctica's ice sheets are mentioned in plural in the title and early in the abstract but there is no definition of what these ice sheets are. The reader must presume this refers to mean East and West Antarctic Ice sheets but this is not made explicit until much later on in the text.

This could be made explicit, or perhaps in the early text either the singular Antarctic Ice Sheet might be better, or perhaps just 'Antarctic ice' with the sheets to be defined later as the distinction becomes important.

We now explain the plural as the abstract opens (Lines 69-70) (it is no longer part of the paper title). We think this should be sufficient to clear up any doubt that by using the plural we are essentially conceptualising Antarctica as being covered by the West Antarctic Ice Sheet and East Antarctic Ice Sheet.

2) Elements of the article read like a list of achievements completed and key findings, but the synthesis of these into a broader understanding sometimes is lacking.

In particular, for section 3 I saw the value its structure with respect to historical perspective, covering differing (yet co-evolving) equipment and data processing etc, it seems a little writer-focused, rather than reader focused. Instead of going through by data providers, I think this section might be rewritten focusing on the main 'eras' of airborne RES surveying, each with different value to resolving architecture.

With some transitional work, the main 'eras' seem to be 1) non-GNSS, analogue, incoherent; 2) GNSS, digital, incoherent; 3) GNSS, digital, coherent....so this would be my suggestion for section 3.1, 3.2, 3.3, with 3.4 for ground surveys

This guidance, with some similar advice from Referee #1, shaped our revision and condensing of the new Sections 3 and 4.

In section 5, to connect the applications to the data better, I would recommend at the beginning of each subsection to include a short introduction to the physical premise of the application...that is, how is the phenomenon expected to be observed in RES data....for example in section 5.4 ice flow dynamics, you might say "Moving ice causes originally flat layers to deform through folding, tilting and disruption. Therefore, deformed isochrons may be analysed to interpret past ice motions"

We have added the following:

At the start of Section 5.1 (Lines 577-578) we have added: "The layering found in ice cores is also visible in radiostratigraphy, as a function of the RES-system resolution (Section 2), ..."

At the start of Section 5.2, (Lines 620-621) we have added: "Successive snowfall events create a record of progressively buried isochrones which can be observed in radargrams."

At the start of Section 5.3 (Lines 647-648), we have revised the wording to: "The presence of a subglacial water body or enhanced geothermal heat flux draws isochrones down towards the ice base. Exploiting this principle, isochrones have been used to calculate melting at the base of the ice."

At the start of Section 5.4 (Lines 673-674), we have added: "Moving ice causes IRHs that were originally deposited flat at the surface to deform through folding, tilting and disruption. Therefore, deformed isochrones may be analysed to interpret past ice-flow dynamics."

At the start of Section 5.5 (Lines 729-730), we have added: "Ice located near to the bed of an ice sheet is typically expected to have undergone strong deformation due to shear, or to originate from processes other than earlier surface accumulation."

3) Section ordering was not ideal for me to follow the concepts

For me, section 4 I think is logically precedent to section 3 if we consider the need of the reader to understand the means of measuring and defining ice architecture, before we can meaningfully consider the value of surveys done. At least, before section 3, where we learn a lot of details about different radar systems of different providers, we need some brief introduction to the main factors that impact on data quality...this is all made quite clear in section 4.1 to 4.4 though so they could be moved en-bloc. Section 4.5 could be part of section 5

This guidance is followed in the new Sections 3 and 4.

4) The discussion needs a more rigorous defense

Section 6 outlines the importance of the work to develop deeper and broader knowledge of Antarctica's ice. I felt for this section that while it is a good identification of future directions within

the scope of the review, it lacked a framing in the context of targeted decisions for the optimal outcome.

For section 6.1 My key question is what is the (relative) cost vs value vs risk proposition for each of these given technological, logistical and financial constraints? What is the low-hanging fruit and what are the grand challenges? I would appreciate here a table with some information on what resources each is expected to require (e.g. IP, funding, technology, logistics, infrastructure) to achieve a significant improvement, and perhaps some indication of if these requirements can be met in the next decade or so.

We feel that the exercise of engaging in the language suggested here is beyond the scope of this review paper, but is certainly a logical extension in the forum of SCAR, science programmes and future grant applications. We think the exercise would be appropriate for a "white paper" building on this science review article.

For section 6.2 Deliverable is not the right word for this section. For me, a deliverable is a finite and defined outcome with a specific timeframe. Perhaps outcomes?

We have changed the section title and replaced the word "deliverable" throughout Sections 6.2 and 6.3 (and much earlier at Line 161 when we signpost the paper structure).

In this section it is not very clear to me the mechanisms by which these might be achieved . I think to identify the scale of the collaboration needed (i.e. is it needing a few research groups or a major multinational program), and also look at these through existing collaborative frameworks (are they sufficient?), and also to look at opportunities/needs for broad new collaboration.

Similarly to two suggestions above, we think this is more the job of a "white paper" (ideally coordinated through SCAR) that can be underpinned by the science objectives set out in this paper.

Given the length of the manuscript, to close out, I think we could do with a distinct conclusion, even just 2 paragraphs, to reaffirm the main points of the text.

We have added a new Section 7: Conclusions. The first paragraph is new and does the requested job of reaffirming the main points of the text. However, we would like to hold the principle that the paper finishes with a statement on the equal importance of building a sustainable and diverse community to achieve the science goals, and found the most effective way to do this was to move what had previously been the final paragraph of the former Section 6 to the end of this new overall conclusions section.

**Minor comments**

line 72: what is the basis of these proxy records?

We have added some examples in a parenthesis (Lines 74-75).

line 104: also sediments are often very limited on the timescales of observation, and further are quite indirect with respect to ice conditions

We have added this wording in here (Lines 109-110).

line 106: which ice sheets East and West? Here is a good place to introduce the challenge of a composite ice sheet with several distinct parts.

This is now introduced in the abstract (Lines 69-70). Maybe we could add something here too, but are wary of over-extending the length of the introductory paragraph.

line 107-108 - can you be more specific here?

We are not sure what is being requested here so have taken no action.

line 115 - a point to consider here is the extent to which RES-derived architecture can be considered total architecture...perhaps a brief comment that RES can't resolve everything

We were careful here to phrase this already as "RES-imaged" (Line 120). Here we would like to keep the focus on introducing the term as specifically what RES sees.

Figure 1: Axes need to be 3D to match the figure.

We are ashamed not to have noticed this in the first place! Corrected (Fig. 1).

line 124 - here again is the plural necessary?..if so we need some definition.

Addressed above.

line 160: complex as in complicated or complex as in real and imaginary numbers? perhaps avoid complex

We meant as in real and imaginary numbers but you are correct that the technicality is not needed here and we have excised it (Line 169).

line 194: this active field of research presumably has at least one paper to cite

We have added Castelletti et al. (2021) (Line 203).

line 207: instead of for over, perhaps spanning

This is now the opening of Section 4. We have adopted the suggestion (Line 400).

line 218: 'now commonly employ state of the art' is redundant. In the absence of specifics, I think improved would do

Agreed but our revision of the section has rendered the point moot.

line 225: 'shallow' RES here refers to 100s m where before and in Fig 2 the 'shallow' sounding reached 2 km. Perhaps a different temrinology could be used to differentiate - perhaps shallow vs intermediate vs deep-penetrating RES

Throughout the revised paper we are now careful to use "shallow" to describe imagery that penetrates a maximum 2 km of the ice column, "deep" to describe our main focus (radar systems that see through the full ice column regardless of thickness), and a new term, "near-surface" to deal with imaging the upper few tens of metres.

line 249-250 - I think the brackets can go as they include a full sentence

line 251 - 'from before' replace with preceding

line 252 - 'were' needs replaced with a verb - perhaps 'being' or 'becoming'

line 254 - 'acquiring data digitally' > 'digital data acquisition'

line 280 'positional uncertainty' can be singular I think?

line 321 - delete shallow as the depth is defined numerically

line 323 - 'across both west and Eat Antarctica'. I find this confusing. Are some radar setups east and west specific but PASIN is not? Is there a important difference in acquisition for East vs West Antarctica ... if it is just that surveys have been done in multiple places then I would omit this point

All of the above is now incorporated in the updated Section 4.

line 532 - 'density value ... of the order of several meters'. Unless I misunderstand the use of density this does not make sense

We agree this was not best phrased; we were alluding to the commonly-used "firn depth correction" which is measured in metres. This is now clarified (Line 270).

line 549 - Here it might be useful to distinguish semi-automated and fully-automated...if that is the intent

We have clarified that automated means fully automated (Line 288).

line 560 - My question here is whether IRH brightness necessarily translates to significance for ice conditions. This is addressed later, but perhaps a brief comment here on this link between data and reality is warranted.

We have considered this but since it is covered later in the paper we have decided not to add anything further at this stage.

Figure 8: I found this figure, while quite nice, but very complex and hard to glean any information from. Apart from the impression that work has been done i didn't learn so much. Perhaps fewer sub-figures and more annotations would be better?

We take the point but the figure has already been through multiple iterations through the large authorship team so we have elected not to implement further substantive changes.

line 813 - drawing parallels with structural geology, a technical question is whether the details of the fold geometry can indicate the deformation mechanism, or if it is ambiguous.

We think that the current paragraph (Lines 673-693) just before this sentence (Lines 694-697) opens the next paragraph already details that it is ambiguous, and indeed very challenging to deconvolve which mechanisms between passage through regions of high strain, flow over and around large obstacles, and/or strong variations in ice rheology predominate to cause the folding observed at any given location. Therefore we have not actioned any changed to the previous wording.

Our understanding of this could improve significantly with a greatly enhanced database of Antarctica's internal architecture against which to assess models!

line 851 - winnowed is not I think the right word...perhaps pinched out or truncated?

We have adopted "pinched out" (Line 734).

line 943 - I don't get the sense here of whether you mean any model might do OK or more that for each circumstance there is an optimal model...this needs rewritten to be clear

We mean that certain models are better for certain situations. We have modified the text so that this is clearer (Lines 827-830).

line 1158 - this strikes me that the outcome of this approach would more be an upper limit on heat flux but this does not negate the value

Agreed but this does not require revised wording.

line 1226 - This is a key step. The need also for a standardised and automated processing (as far as it possible) is also important so might be added here if relevant. A key goal I think should be to move away from 'decadal' compilations and more towards and ongoing resource...this needs automation

We treat this as a supportive comment rather than a suggestion that anything further needed to be added to the wording in the manuscript.

We are certainly on board with the vision that the database should grow in an ongoing and accessible manner rather than in decadal compilations. Our challenge is that even before then the standards for lodging data need to be agreed upon.

We are grateful to Dr. Alan Aitken for his time spent reading the manuscript and formulating this considered advice.

**Response to Editor pre-review comments from 9 Sep 2024 (Huw Horgan)**

Para L231-242. Similar paragraphs include representative references. Suggest doing the same here.

L244. Shift definition of coherent/incoherent from later in para to here.

Para L456-469. The use of parentheses and semi colons make this para difficult to digest. Suggest rewrite.

Fig 4c) "differentation" -> differentiation?

All four comments above actioned via the Section 3/4 overhaul.

Fig 9. I appreciate this 3-dimensional information is hard to communicate. Perhaps a combination of persepective and 2D distance-depth slices? Also, it looks like there are some velocity vectors not mentioned in caption.

Actioned (in a minor way) as per response to Referee #1 comment on Fig. 9 above (p.2 of response document).

Section 6.1.4 would follow more logically after 6.1.1 or they could be combined.

Our logic with the existing order is that modelling is needed to guide where data most need to be analysed, whether they exist already (as dealt with by Sect. 6.1.2 and 6.1.3) or requiring new data (Sect. 6.1.4). Therefore we have not adopted this suggestion.

L1046 I expected a RINGS reference here, similar to L1193.

On Line 1193 (now Line 1093) the complement with SCAR AntVolc is self-evident. For the context of Line 1046 (now 929) a reason to mention RINGS is not so clear. The point we make here is about identifying regions all the way from the coast to 100s of km inland that are not surveyed yet with digital systems. RINGS has a specific focus on the coastal regions and will not motivate surveys that fill these gaps.

L1129 The definition of shallow radiostratigraphy could come earlier in the paper.

In the revision we have re-termed this near-surface radiostratigraphy to distinguish it from shallow radiostratigraphy (upper few 10s of m vs up to 2 km depth). Now near-surface is introduced at Line

292, shallow is introduced at Line 468, and the wording that used to be at Line 1129 and is now Lines 1026-1028 is altered to include a reminder of our definition of near-surface, since it comes so much later in the paper.

L1135 '...should be a "shallow" of pan' -> should be a review of...

This now correctly reads '...should be the development of a "near-surface pan-...' (Lines 1034-1035).

Appendix suggest WGS84 elevations and a.s.l elevations. If a.s.l a geoid must also be referenced in the data structure. Suggest WGS84 ellipsoid reference datum or designate space for geoid.

We have changed to WGS84 ellipsoid reference datum (Table A1).

**Response to Editor post-review comments from (Huw Horgan)**

In your editor decision statement on 8 April 2025, you were kind enough to detail a number of your thoughts on specific points raised by the reviews. It was helpful to have this added perspective, thank you; in practice, all the points raised in your overview have already been incorporated into the above.